

# Metamorphism of Arctic marine snow during the melt season. Impact on spectral albedo and radiative fluxes through snow

Gauthier Vérin[1,2], Florent Domine[1,3], Marcel Babin[1], Ghislain Picard[2], Laurent Arnaud[2]

[1] Takuvik Joint International Laboratory, Université Laval (Canada) and CNRS-INSU (France), Université Laval, Québec, Canada
[2] Univ. Grenoble Alpes, CNRS, Institut des Géosciences de l'Environnement (IGE), UMR 5001, Grenoble, 38041, France
[3] Department of Chemistry and Centre for Northern Studies, Université Laval, Québec, Canada

*Correspondence to*: Florent Domine (florent.domine@gmail.com)

**Abstract.** The energy budget of Arctic sea ice is strongly affected by the snow cover. Intensive sampling of snow properties was conducted near Qikiqtarjuak in Baffin Bay on typical landfast sea ice during two melt seasons in 2015 and 2016. The sampling included stratigraphy, vertical profiles of snow specific surface area (SSA), density and irradiance, and spectral albedo (300-1100 nm). Both years featured four main phases: I) dry snow cover, II) surface melting, III) ripe snowpack and IV) melt pond formation. Each phase was characterized by distinctive physical and optical properties. A high SSA value of 49.3 $m^2\,kg^{-1}$ was measured during phase I on surface wind slabs together with a corresponding broadband albedo of 0.87. Phase II was marked by alternating episodes of surface melting which dramatically decreased the SSA below 3 $m^2\,kg^{-1}$ and episodes of snowfall reestablishing pre-melt conditions. Albedo was highly time-variable with minimum values at 1000 nm around 0.45. In Phase III, continued melting led to a fully ripe snowpack composed of clustered rounded grains. Albedo began to decrease in the visible as snow thickness decreased but remained steady at longer wavelengths. Moreover, significant spatial variability appeared for the first time following snow depth heterogeneity. Spectral albedo was simulated by radiative transfer using measured SSA and density vertical profile, and impurity contents based on measurements. Simulations were most of the time within 1% of measurements in the visible and within 2% in the infrared. Simulations allowed the calculation of albedo and of the spectral flux at the top of the sea ice. These showed that photosynthetically active radiation fluxes at the bottom of the snowpack durably exceeded 5 $W\,m^{-2}$ (~9.2 $\mu mol\,m^{-2}\,s^{-1}$) only when the snowpack thickness started to decrease at the end of Phase II.



## 1 Introduction

Sea ice features and dynamics in the Arctic are undergoing radical changes, including a shift from multi- to first-year ice (Comiso, 2012), a decrease in thickness (Kwok and Rothrock, 2009) and in September areal extent (Comiso et al., 2017), and an earlier break-up and a later freeze-up (Arrigo and van Dijken, 2011). These changes strongly affect air-sea interactions (momentum, heat, gases) with multiple feedback loops (Serreze and Barry, 2011; Stroeve et al., 2012). They also affect marine ecosystems by substantially increasing the amount of sunlight that penetrates into the ocean and supports

photosynthesis under sea ice, and in the open ocean during the now longer ice-free-season (Ardyna and Arrigo, 2020).

The snow cover plays a significant role in Arctic sea ice evolution. Indeed, during winter and early spring, dry snow reflects up to 90% of incoming solar radiation which drastically reduces the energy absorbed by the underlying sea ice (Grenfell and Maykut, 1977; Grenfell and Perovich, 2004; Nicolaus et al., 2010). Snow delays the onset of sea ice melt while its albedo remains high enough, and thus directly drives the duration of the melt season which is itself related to the

minimum sea ice extent in September (Perovich et al., 2007). The snow optical properties also control the amount of light reaching the upper ocean column under sea ice, much more so than sea ice itself (Grenfell and Maykut, 1977; Perovich, 1990). It was recently shown that major phytoplankton blooms can take place under sea ice and that snow is the main driver of bloom onset (Ardyna et al., 2020).

The albedo of snow depends mostly on the optical grain size, the density, the snowpack thickness and the impurity content

of the snow (Aoki et al., 2003; Warren, 1982). Snow is a highly scattering medium composed of ice particles that are weakly absorbing in the visible range (Picard et al., 2016a). The albedo increases when the snow particle size decreases, and these changes mostly take place in the infrared. A snow grain size metric relevant to optical studies is the optical diameter $d_{opt}$, i.e. the diameter of spheres having the same surface area to volume ratio as the snow (Grenfell and Warren, 1999). At present this is obtained by measuring the snow specific surface area (SSA, in $m^2$ $kg^{-1}$). Both are linked with the

relationship SSA=$6/\rho_{ice} d_{opt}$, with $\rho_{ice}$ the density of ice. (Domine et al., 2006) showed that SSA was well related to snow reflectance in the infrared range, and SSA measurements using reflectance at 1310 nm are now routinely used (Gallet et al., 2011; Gallet et al., 2009). Despite the great use of snow SSA, its measurements remain scarce on continental snowpacks, and even more so on sea ice where they are limited to a few profiles (Domine et al., 2002; Domine et al., 2012), limiting our understanding of the albedo of snow on sea ice.

Once deposited on sea ice, snow grains undergo continuous transformations over time known as snow metamorphism, which is mostly driven by meteorological conditions. While the snowpack is dry, the main factors responsible for metamorphism are the temperature gradient in the snowpack and wind. The temperature gradient between the warmer sea ice and the colder atmosphere leads to an upward water vapor flux coupled to sublimation/condensation cycles that lead to grain growth and the formation of faceted crystals and ultimately large hollow depth hoar crystals (Colbeck, 1983). The

upward vapor flux also leads to mass loss in the basal layers so that the density of depth hoar layers often decreases over the season (Domine et al., 2016b). Wind, on the contrary, leads to snow drifting and to the sublimation and fragmentation of grains so that wind processes produce hard dense wind slabs made of small and mostly rounded grains. Since the temperature gradient is greatest during the beginning of the snow season when the snowpack is thin and the surface of the thin sea ice is still warm, depth hoar formation is more likely then. Arctic snowpacks thus usually feature basal depth hoar

layers topped by wind slabs and occasionally fresh snow before it gets wind-blown (Domine et al., 2016b; Sturm et al., 2002). Overall snow thickness ranges from a few centimeters up to 70 cm with densities ranging from 70 to 500 kg $m^{-3}$ (Domine et al., 2002; Domine et al., 2012; Gallet et al., 2017; Sturm et al., 2002). The thickest snowpacks are often found



on highly deformed sea ice where features like ridges trap wind-drifted snow (Sturm et al., 2002). When melting starts in spring, snow grains become rounded and daily freeze/thaw cycles leads to rapid grain growth and to the formation of hard dense refrozen layers made of large rounded grains (Colbeck, 1973). In general, snow metamorphism leads to decreases in SSA (Legagneux and Domine, 2005) and consequently in albedo (Domine et al., 2006). Snowpack properties therefore vary over time. Given the large wind-induced spatial heterogeneity (Filhol and Sturm, 2015), the snowpack on sea ice shows large time and space variability which makes the field study of snow properties and in particular albedo challenging, because they require a lot of samplings over a relevant time period.

The snowmelt period leads to major and sudden changes over sea ice. It extends from the first surface melt event to the formation of melt ponds with typical durations ranging from 10 days to one month (Perovich et al., 2007; Perovich and Polashenski, 2012; Sturm et al., 2002). It can be triggered by weather conditions such as positive temperatures or rain events (Nicolaus et al., 2010; Sturm et al., 2002). Surface melting results in the formation of a thin surface layer of rounded grains which tends to thicken with further melting. Once wet metamorphism reaches the bottom of the snowpack, the remaining snow layers melt rapidly (Gallet et al., 2017). As snow grains grow and SSA decreases, albedo drops remarkably, almost doubling the radiative energy absorption (Nicolaus et al., 2010; Perovich et al., 2002). This acts as a positive feedback enhancing further melting. The combined effects of surface melting and atmosphere warming enhance the air moisture content, often producing persistent overcast conditions leading to snow precipitations (Maksimovich and Vihma, 2012; Mortin et al., 2016). (Perovich et al., 2002), (Gallet et al., 2017) and (Perovich et al., 2017) observed sudden increases in albedo after such fresh snow precipitations which suddenly increased the snow albedo and interrupted melt progression. Furthermore, sufficient summer snowfalls occasionally allow the snowpack to persist through the entire summer (Warren et al., 1999). Melt onset appears to be a chaotic transient period on the Arctic ocean, as changing meteorology can significantly lengthen the melting period.

Over the last decades, considerable effort has been made to better understand the radiative properties of snow on sea ice and their evolutions across seasons. Snow albedo drives processes which control the energy budget of sea ice and the radiative flux to the underlying ocean, and albedo itself depends on snow properties. However, studies which aim to link physical and optical properties of snow still remain largely qualitative. Today, data are lacking to fully quantify and model the global radiative transfer of sea ice because we do not have time series of the snow properties required to understand albedo evolution. Moreover, present data sets do not include systematic combined measurements of snow optical and physical properties at the same spot. This lack of data is particularly detrimental during the melting period when albedo is highly time-variable following alternation of freezing and melting events and precipitation.

The purpose of this paper is to contribute to filling these gaps. We simultaneously documented the temporal evolution of snow physical properties and albedo during two melting periods (2015 and 2016) on typical Arctic landfast sea ice on the east coast of Baffin Island. One or two snowpits were sampled almost every day. Measurements included, for the first time over sea ice, the time evolution of the SSA vertical profile, and the corresponding spectral albedo. The stratigraphy and snow density vertical profile were also documented. In addition, the optical absorption caused by light-absorbing particles (LAPs) was determined using vertical irradiance profiles in sufficiently thick snowpacks and using measurements on melted snow samples. We first aim at linking surface conditions characterized by snow physical properties (SSA and density) and impurity content with albedo. The second objective is to verify that measurements of vertical profiles of snow properties enable reliable simulations of the albedo of snow-covered sea ice, especially during the chaotic melting period. Finally, we briefly compare simulated radiative fluxes at the snow-ice interface with published underwater measurements (Massicotte et al., 2020) of photosynthetically active radiation and chlorophyll *a* concentration to evaluate the potential of



snow physical modeling in determining phytoplankton blooms under sea ice.

**2 Materials and methods**

**2.1 Study area**

Field sampling was conducted close to Qikiqtarjuak Island in Baffin Bay (Figure 1) from May 12 to June 18 in 2015 and from May 17 to June 25 in 2016 as part of the Green Edge project (Massicotte et al., 2020). All measurements were made on typical landfast sea ice a few hundred meters around an ice camp (same location for both years). In 2015 the ice was very smooth whereas in 2016 the ice surface was deformed with small reliefs and ridges because of an early break up in December 2015. The sampling periods were selected to capture the melting stage but complete snow melt-out and pond formations were only observed in 2016. A meteorological station was setup close to the ice camp and provided continuous measurements of 2-m air temperature and of snow thickness (Massicotte et al., 2020).

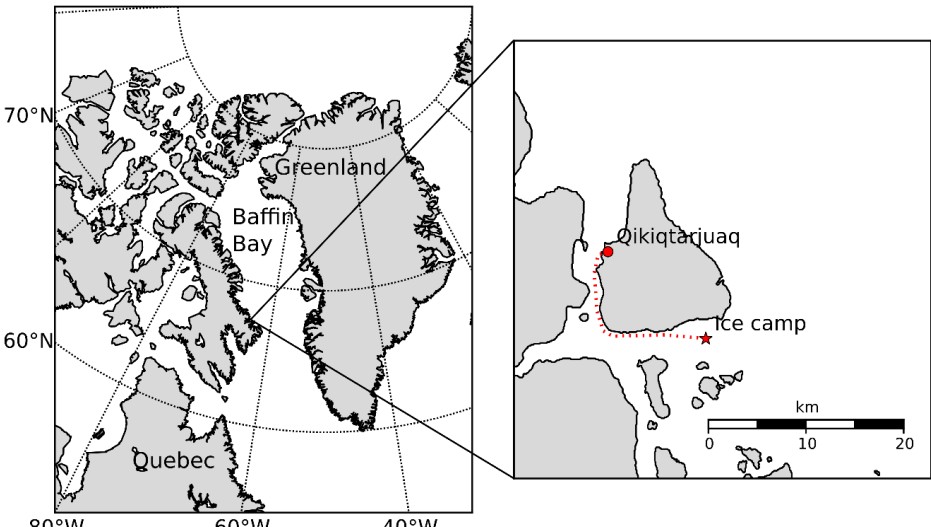

**Figure 1. Location of the measurement site close to Qikiqtarjuaq island (67° 33' 29" N, 64° 01' 29" W), east coast of Baffin Island, Canada.**

**2.2 Optical properties**

**2.2.1 Albedo measurements**

Albedo measurements were performed with a custom-built radiometer (Solalb, developed at IGE following (Belke-Brea et al., 2020; Picard et al., 2016b)). Light was collected using a cosine collector and guided through an optical fiber to a spectrometer (Maya 2000 PRO, Ocean Optics). Irradiance was measured over the 300 to 1100 nm wavelength range with 3 nm resolution. More details about the cosine collector are given in (Picard et al., 2016b). The cosine was fixed at the end of a 3-meter aluminum pole which rested on a tripod 70 cm above the surface (Figure 2B). At the other end, the operator manually controlled the arm and triggered the spectrometer. The distance between the operator and the cosine collector (Figure 2B) ensures minimal disturbance by the shadow. The horizontality was ensured by the operator within less than 0.3° using an electronic two-axis inclinometer mounted near the cosine collector. Albedo determinations required the measurements of upwelling and downwelling irradiance, which were made sequentially using the same cosine collector



with the pole being manually rotated by 180°. No absolute or relative calibration was needed, but measurements had to be made under steady incident light conditions during the 30 s process, which seldom strictly prevailed during the Arctic spring. The setup therefore included a reference photodiode to measure light fluctuations at all times for subsequent

correction. For both upwelling and downwelling irradiance measurements, the integration times was automatically adjusted in order to optimize the signal to noise ratio. A single operator could manage the entire process including albedo measurements along linear transects.

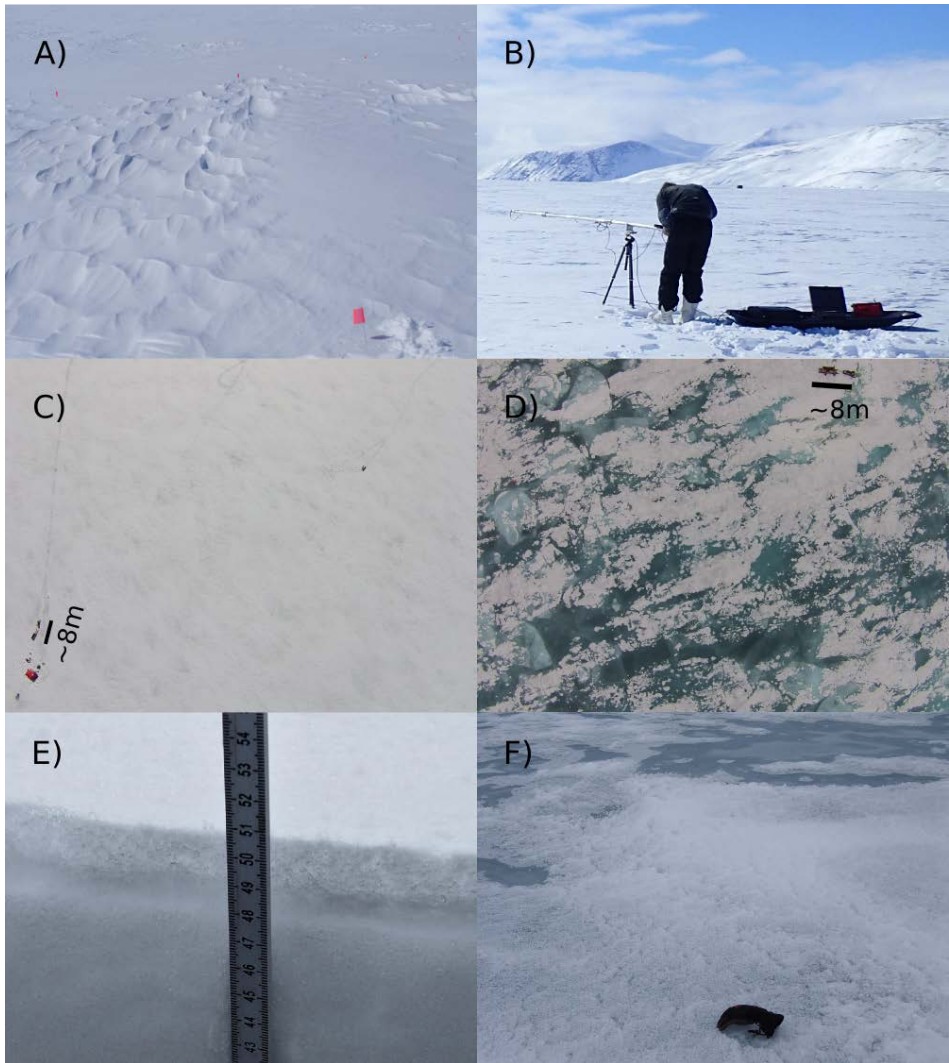

**Figure 2. Pictures of sea ice at different stages of the 2015 and 2016 melting season : A) dry snow cover, 26 May 2016, B) albedo measurement with SOLALB (2015/05/19), C) snow melting, spatial variability begins to be observable (drone picture 2016/06/13), D) melt pond formation (drone picture 2016/06/25), E) Typical snow surface in phase II with a thin layer of fresh snow covering a 1.5 cm thick layer of wet grains (2016/06/04 sp1) and F) picture of refrozen melt ponds on June 18 (2016) covered by a thin layer of fresh snow ; glove for scale.**



### 2.2.2 Irradiance profiles

Vertical irradiance profiles in the snow were performed using the SOLEXS radiometer detailed in (Picard et al., 2016a). Snow dunes were selected for these measurements because thicker layers are more suitable to determine accurately irradiance profiles. The optical absorption by LAPs in the snowpack were determined from the exponential rate of irradiance decrease in a homogeneous layer (Belke-Brea et al., 2021; Tuzet et al., 2019) so that thicker layers yield more reliable measurements. Another reason for selecting thick snowpacks is that irradiance data in the top 7 cm of the profile had to be discarded because of near-surface perturbations (Belke-Brea et al., 2021; Picard et al., 2016a). Briefly, an optical fiber was slowly lowered through a hole preformed in the snowpack. The fiber was guided by a vertical rail mounted on a four-legged structure (Picard et al., 2016a). A profile requires at least one minute to complete, which was a major issue under changing light conditions. These were monitored by the same photodiode as for albedo measurements. Measurements were repeated until a complete profile could be recorded under constant incident light. Two such valid profiles were recorded for each snowpit. They were averaged and then interpolated over 5-mm intervals. All final profiles were normalized to the irradiance at a depth of 7 cm (Picard et al., 2016a). Analysis of the profiles yield the spectral absorbance of LAPs, which can be used to determine the type of absorbing particles (essentially black carbon (BC) or mineral dust). LAPs were also analyzed using chemical methods, as detailed in the following section.

### 2.2.3 Light-absorbing particles in snow

Absorption spectra of light-absorbing particles in snow were measured during the 2016 field campaign. Twelve snow samples (500 ml each) were collected at random locations around the ice camp before complete melt out. Samples were brought to the lab where they were melted within a closed cooler placed at ambient indoor temperature. For particulate absorption, we followed the so-called T-R method developed by (Tassan and Ferrari, 1995, 2002). Briefly, water samples were filtered under low vacuum on Whatman GF/F glass fiber filters immediately after snow melting. Optical density (dimensionless) of all particles on these filters, referred to as $OD_p(\lambda)$, was then determined from transmittance and reflectance measurements from 200 to 800 nm with 1-nm resolution using a Cary 100 spectrophotometer (Agilent Technologies). A baseline correction was applied by subtracting the optical density of a fully hydrated blank filter from $OD_p(\lambda)$. A correction for pathlength amplification was applied as proposed by (Tassan and Ferrari, 2002). We then converted these $OD_p$ values into absorption coefficients for total particles ($a_p(\lambda)$, m$^{-1}$) using the following equation:

$$a_p(\lambda) = 2.3 \, OD_p(\lambda) \, \frac{FA}{V} \qquad\qquad (1),$$

where $FA$ is the filtration area on the glass fiber filter (m$^2$), and $V$ is the volume of the melted snow sample (m$^3$). The 2.3 factor accounts for the conversion from base 10 (i.e. $OD_p$) to base $e$ (i.e. $a_p$) logarithm.

### 2.3 Snow physical properties

Here, snow physical properties refer to temperature, snow grain shape and visual size, SSA and snow density in each snowpit. We first identified the main stratigraphic layers by visual inspection. For each layer, the average visual snow grain size and shape were determined using a hand lens. Snow temperature was measured with a Pt100 thermistor probe at several depths from the bottom of the cover to up to 10 cm beneath the surface. Negative freeboard was reported when the sea level was above the snow-ice interface. The vertical profile of snow density was measured using a 100 cm$^3$, 3 cm high box cutter. The collected snow sample was weighed using an electronic scale. According to (Conger and McClung, 2009), this method allows snow density measurements with an accuracy of 11%. The main uncertainties concern the real





volume extracted by the cutter, which depends on the type of snow. Finally, vertical profiles of SSA were determined from the snow IR reflectance using the DUFISSS instrument (Gallet et al., 2009). Briefly, DUFISSS measures the albedo of a cylindrical snow sample 63 mm in diameter and 25 mm thick at 1310 nm with an integrating sphere. The SSA is deduced

from the albedo using a polynomial relationship. The correction concerning the determination of SSA of wet snow introduced by (Gallet et al., 2014a), which mostly consists in adding 0,5 $m^2$ $kg^{-1}$ to the value of SSA, was not applied in this study because it did not induce significant changes in albedo simulations (less than 1% albedo increase at 1000 nm in most cases and negligible change in the visible). The uncertainty in SSA determinations is 12% under most conditions (Gallet et al., 2009). Melting can occur if the sample is not handled fast enough under warm meteorological conditions,

which leads to a lowered SSA value. Special care was taken to keep sampling tools as cold as possible, e.g. by cooling instruments in bottom snow layers when the surface was melting.

### 2.4 Sampling Protocol

Data presented in this study were collected either in snowpits or along transects.

*Snowpits:* Albedo was measured first since it requires a pristine area. A minimum of 3 measurements were made depending

on sky conditions and light variations. All of them were performed facing the sun to avoid any shadow from the operator and the equipment. Irradiance profiles were them measured when thick snowpacks were studied, with minimal perturbation to the snow surface. When irradiance profiles were performed, the pit was then dug so as to intersect the hole made for the irradiance profile. All stratigraphic measurements were carried out along a one-meter-long shaded trench. Our objective was to conduct all samplings at the same place in order to fully characterize physical and optical properties of the snow at

each station.

One or two snowpits (each requiring three hours of work, more for thick snowpacks) were sampled each sampling day. Fewer snowpits were sampled in 2016 (10 versus 35 in 2015) because the snowpack was already ripe (i.e. isothermal at 0°C and melting throughout) before sampling operations. Snowpit locations were randomly chosen around the ice camp. Particular efforts were made to sample the widest possible range of snowpack depths in order to address spatial variability.

*Transects:* Albedo and snow depth were also measured every 5 m along transects (100 m to 150 m long). All the equipment was placed on a sled to facilitate transport. These transects were performed in 2016 only. As the snowpack was already ripe, the study of spatial variability using large scale measurements was favored.

### 2.5 Data processing

Upwelling and downwelling irradiance raw acquisitions require several processing steps before the albedo can be obtained.

During the field campaigns, spectra were visually checked at the end of the sampling day. Unrealistic data, based on qualitative criteria (spectra shape, range of values, signal to noise ratio), were rejected. The first step of processing was to remove the systematic offset in both acquisitions caused by dark current and stray light effects. This offset was approximated for each acquisition as the mean signal at short wavelength (between 200 nm and 260 nm), because there is no incoming photon in this wavelength range. Dark current was assumed to be constant over the entire wavelength range.

Then, spectra were divided by their corresponding integration time. Our cosine collectors had been previously characterized on an optical bench in order to assess their exact angular response (Picard et al., 2016b). This angular response was then used to correct the downwelling irradiance measurements depending on the solar zenith angle (SZA) during the acquisition if the sky was clear. No correction was needed for overcast skies. We excluded any acquisition for which the reference photodiode signal varied by more than 2% between the upwelling and downwelling irradiance



measurements. Below 2%, spectra were rescaled using the reference photodiode signal assuming that changes in incident light were equivalent over the entire wavelength range. After all these steps, albedo was calculated as the ratio of upwelling over downwelling irradiance. Albedo spectra were finally smoothed using a low-pass filter in order to eliminate the noise inherent in the measurements. For each measurement site, it was checked that all spectra correctly overlapped before being averaged. For the 2015 dataset, the average standard deviation of all integrated albedos (over the 400-1000 nm wavelength

range) measured at each snowpit is 0.3% with a maximum of 1%. Thus, in most cases, it is reasonable to assume that the precision in albedo measurements is better than 1%. This uncertainty does not account for external error sources such as varying illumination conditions or unexpected shadows or reflections.

**2.6 Radiative transfer modeling**

The numerical simulations of albedo and irradiance profiles were performed using the Two-stream Analytical Radiative

TransfEr In Snow (TARTES) model (Libois et al., 2013). Briefly, TARTES uses the delta Edington approximation (Jimenez-Aquino and Varela, 2005) in a layered plane-parallel snowpack. Each layer is characterized by an average SSA, density and impurity concentrations. Following the asymptotic radiative transfer theory (ART) (Kokhanovsky and Zege, 2004), light scattering in snow is also determined by the absorption enhancement parameter B, which characterizes multiple reflections within a snow grain, and by the asymmetry factor g, which describes the preferential scattering

direction (e.g. forward or backward).

TARTES solves the radiative transfer equation at all depths. The underlying sea ice is not modeled, only its albedo (measured in the field) is specified at the bottom of the snowpacks. The snow albedo depends on SZA and cloud cover, but a fully diffuse radiation is equivalent to a direct radiation with SZA ~50° (Warren, 1982). In our study, SZAs were between 47° and 57°, therefore simulations were performed considering diffuse radiation (SZA of 53° in TARTES, according to ART). Doing so, the maximal error on albedo is ~0.01 at 1000 nm.


The use of TARTES allows the calculation of albedo on a wide wavelength range which makes possible the assessment of the broadband albedo $\alpha$ and total energy absorbed at the snow surface A, in W m$^{-2}$. Both were calculated as follows:

$$\alpha = \int_{300}^{3000} \alpha_S(\lambda) I(\lambda) d\lambda / \int_{300}^{3000} I(\lambda) d\lambda \qquad (2)$$

$$A = \int_{300}^{3000} (1 - \alpha_s(\lambda)) I(\lambda) d\lambda \qquad (3)$$

where $\alpha_s$ is the spectral albedo calculated with TARTES over the 300-3000 nm wavelength range and $I(\lambda)$ is the spectral solar irradiance in W m$^{-2}$ nm$^{-1}$. The solar irradiance spectra was calculated with the SBDART model (Ricchiazzi et al., 1998) for June 1st at 12:00 at Qikiqtarjuaq under clear-sky conditions of Arctic spring on snow-covered areas. Only one solar spectrum was used since the study focused on the broadband albedo more than on the energy budget.

Impurities in TARTES are assumed to be external to the ice grains and the particle size is small compared to the wavelength. Their absorption was taken into account as detailed in (Kokhanovsky and Zege, 2004). TARTES is therefore able to consider any kind of LAP whose refractive index is known (Tuzet et al., 2019). Concentrations and types can be adjusted for each layer of the snowpack.

Finally, since snow layers near the surface and down to a depth of at least 10 cm were almost always windpacks or clustered

rounded grains, we follow the recommendation of (Libois et al., 2014) and use B=1.25 and g=0.895.

**3 Results**



In this section, the 2015 dataset is entirely presented with all the 35 snowpits which comprise vertical profiles of SSA and density, as well as albedo measurements. Only the 2015 data were used for radiative transfer modeling with TARTES. The 2016 dataset is only composed of albedo measurements and qualitative observations on snow and ice. It will be specified in the text whenever we refer to the 2016 dataset.


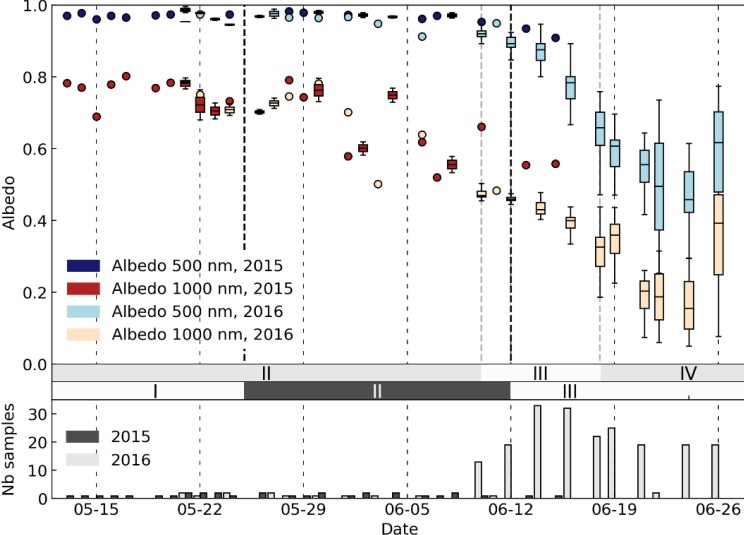

**Figure 3. Time evolution of albedo at 500 nm in blue and 1000 nm in red for the 2015 and 2016 field campaigns in darker and lighter colors respectively. Boxes are used if more than 1 measurement per day is available, colored dots otherwise. The main phases are specified by horizontal grey bars for both years (Phases I, II and III for 2015 and Phases II, III and IV for 2016). Gray bar graphs at the bottom represent the number of albedo measurements per day.**


### 3.1 General evolution and meteorological conditions

Surface conditions changed drastically during both sampling campaigns as depicted in Figure 2. About one month elapsed between the first day of surface melting and meltout. As previously observed by (Perovich et al., 2002) and by (Nicolaus et al., 2010), as the melting season progressed the sea ice surface became darker and spatial variability increased. The time-evolution of albedo at 500 nm and 1000 nm are presented in Figure 3 and, similarly to (Perovich et al., 2002) and (Nicolaus et al., 2010), this evolution clearly shows four main stages confirming visual observations in the field. These phases are defined below.


*Phase I. Cold, dry snow (from the first sampling day on May 13 to May 24 in 2015).* Sea ice was covered by a dry winter snowpack never subjected to melting. Air temperature increased during this phase but remained below 0°C (Figure 4). A significant snowfall event associated with strong winds occurred before the first sampling day in 2015 (May 8 and 9), building a fresh snow layer at least 10 cm-thick. Temperature in snow was first colder at the surface, or at least at mid-depth, (-6.5°C) than at the bottom-most layer where temperatures remained fairly steady between -5°C and -4.5°C in the day time. The subsequent increase in air temperature reversed the temperature gradient in the snowpack during this first phase.



*Phase II. Surface melting (May 25 to June 11 in 2015; from the first sampling day on May 19 to June 9 in 2016).* This phase started with the first surface melting event which coincided with the first positive air temperature in 2015 (Figure





4). Coarse rounded grains and wet grains appeared and albedo decreased in the infrared (Figure 3). Air temperature fluctuated around 0°C and several snowfalls were observed both years during that period (Figure 4, snowfalls specified only for 2015). Moreover, the weather was cloudier than during the previous phase and thick fogs were more common in the early morning. These meteorological conditions persisted in the next phases. Overall, snow temperatures gradually increased until the 0°C isothermal state was reached on June 10, 2015.

*Phase III. Ripe snowpack (June 12 to June 18 in 2015; June 10 to June 17 in 2016).* At this stage, all the snow layers were at the melting temperature and were entirely comprised of rounded polycrystals. This phase is characterized by a decrease in albedo over the visible range for the first time of the season (Figure 3). Snowpack thickness decreased very quickly until melt-out (6 days in 2015, 7 days in 2016). In 2015, measurements were stopped on June 17 so that the end of this phase was not studied. The first melt ponds appeared on June 19, 2015.

*Phase IV. Melt pond formation (in 2016: June 18 to the last sampling day on June 26).* Snowpacks gave way to a mixture of bare ice and melt ponds. The transition between snow cover and bare ice was progressive, because the ice surface was granular and looked similar to the large wet grains observed on the snowpack in the ultimate stages of snow melt. As previously observed, sea ice was first rapidly flooded by extended shallow ponds before they partially drained and got their final shape. During our last sampling day in 2016, June 25, a cooling event associated with snowfall temporally froze the ponds (Figure 2F) and increased albedo (Figure 3).

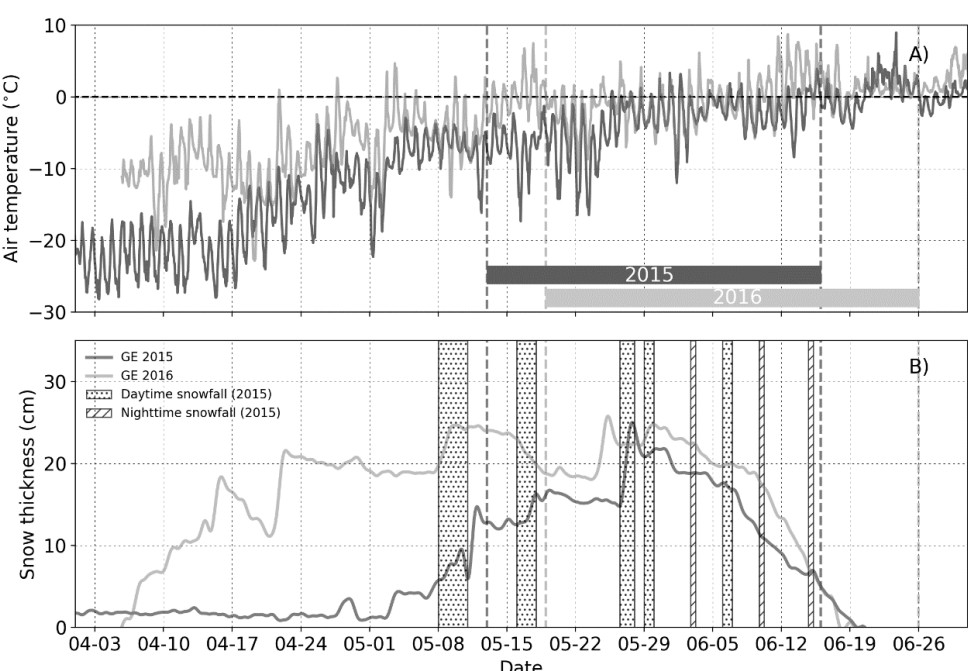

**Figure 4. Continuous measurements of the sea ice meteorological station. Details are given in Massicotte et al. (2020) A) Time-evolution of air temperature and; B) snow thickness measured for Green Edge 2015 (dark gray) and 2016 (light gray). The zero-level snow depth is that measured just after meltout. Gray horizontal bars in (A) denote the sampling periods for both campaigns. Additionally, the main snowfalls in 2015 are specified in (B) with a distinction between conventional snowfalls and light nighttime snowfalls observed at the end of the season.**





### 3.2 Snow stratigraphy and physical properties

Only physical properties sampled in 2015 are presented here because they cover the main first three phases, unlike in 2016. Figure 5 presents the main stratigraphic layers observed for each phase. Regarding the physical properties, vertical profiles of SSA and density are presented in Figure 6 with average values in Table 1.

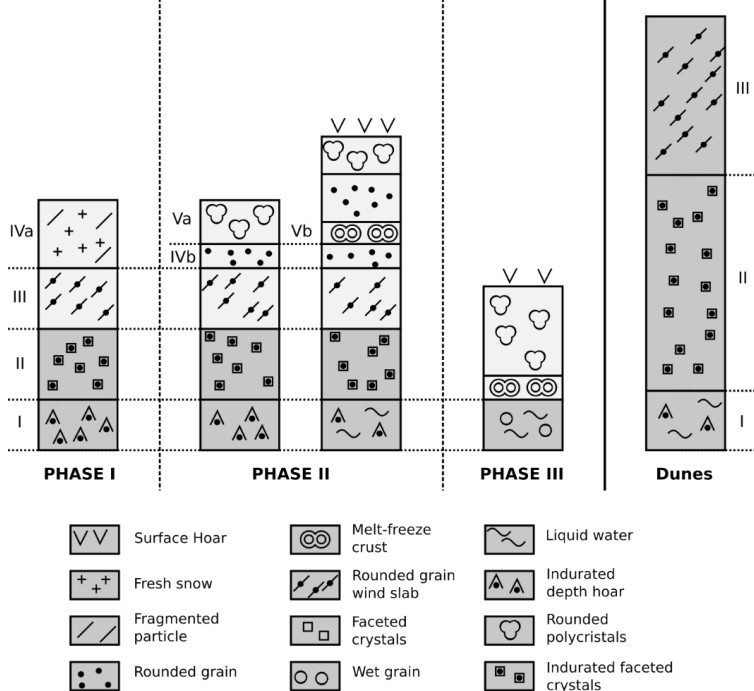

**Figure 5. Main stratigraphic layers observed for each phase and for snow dunes in 2015. The lower layers with low SSA are represented in dark grey. The upper layers with high SSA during phase I are in light grey Layer numbers are those used in Table 1. A vertical scale is not provided as snow depths were highly variable.**

*Phase I. Cold, dry snow.* The observation of 15 snowpits during this phase revealed a dominant stratigraphy composed of three or four main layers. The bottom-most layer (layer I in Figure 5), in contact with the underlying sea ice, was indurated depth hoar formed from a wind slab (Derksen et al., 2009; Domine et al., 2016b), as evidenced by the presence of depth hoar crystals embedded in a matrix of small rounded grains, and confirmed by its high density of $372 \pm 51$ kg m$^{-3}$. Its SSA was $8.9 \pm 4.4$ m$^2$ kg$^{-1}$, typical of depth hoar, whether indurated or not (Domine et al., 2016b). Above this basal layer, a layer of indurated faceted grains (layer II) with average SSA of $12.1 \pm 1.8$ m$^2$ kg$^{-1}$ and average density of $409 \pm 40$ kg m$^{-3}$ was observed. The upper part of the snowpack was first comprised of a wind slab layer (layer III) made of small rounded grains characterized by significantly higher SSA values, $33.4 \pm 2.6$ m$^2$ kg$^{-1}$ and lower densities, $276 \pm 38$ kgm$^{-3}$. Occasionally a layer of dendritic crystals or decomposing particles could be observed on the surface (layer IVa). The highest SSA values were recorded in this layer, $49.3 \pm 5.9$ m$^2$ kg$^{-1}$ (see dark red areas at the surface in Figure 6A). Moreover, sublimation crystals (Gallet et al., 2014b) sometimes formed at the surface of the snowpacks when an upward flux of water vapor condensed into colder air at the snow surface. Figure 6 also shows a significant contrast in both vertical profiles, with layers I and II characterized by lower SSA and higher density than layers III and IV. Moreover, SSA in layer III gradually decreased over time. Overall, snow depth ranged from 15 cm to 54 cm. Snow dunes were studied on May 19,
22, 23, 29 and June 4. They corresponded to the thicker snowpacks. Their stratigraphy revealed a similar sequence as elsewhere, with depth hoar, faceted crystals and wind slab. However, the wind slabs had a much lower SSA than elsewhere, suggesting an earlier formation. The highest densities, reaching 500 kg m$^{-3}$, were measured in layers II of dunes. (Figure 6). Smaller features such as sastrugi (Figure 2a) and barchan dunes were commonly observed before melt onset. Freeboard was always positive during phase I.

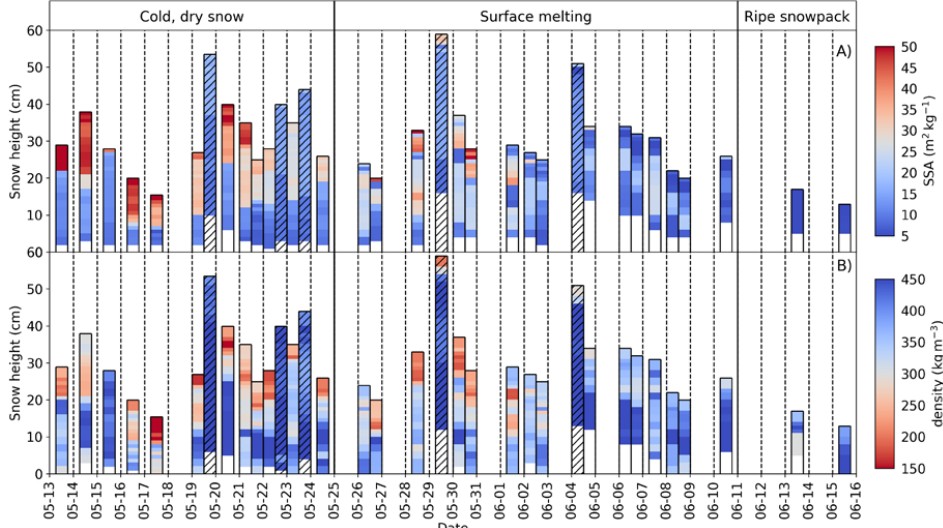

**Figure 6. Vertical profiles of SSA (A) and density (B) for each snowpit sampled in 2015 with snow elevation (in centimeters) on y axis and sampling dates on x axis. Snow dunes are labeled with hatches and phase I to III are specified.**

*Phase II. Surface melting.* The first melting was observed on May 26, one centimeter below the surface, and coincided with a low SSA layer at that depth (Figure 6). Overall, surface melting was characterized by the formation of a layer of rounded polycrystals (layer Va, Figure 5), of low SSA (10.6 ± 4,1 m$^2$ kg$^{-1}$). Additionally, as melting conditions persisted this layer got thicker and its SSA kept decreasing to a minimum of 2.6 m$^2$ kg$^{-1}$ on June 13 (phase III). The alternation of negative and positive temperatures during night and day subjected the surface of the snowpack to a diurnal cycle. During daytime, at the surface, bonds between snow grains melted leading to the observation of wet clustered rounded grain which partially (at least near the surface) froze during the following night, forming again dry rounded polycrystals and often a thin melt-freeze crust at the surface. Several snowfalls (May 16, 17, 26, 27, 29 and June 6, Figure 4) deposited fresh snow layers covering layer Va, which then quickly metamorphized (See the second stratigraphic profile of phase II, Figure 5). Some of these new layers were thick enough to be distinguishable in Figure 6. Fresh snow tended to accumulate in depressions rather than on top of dunes. Melting and subsequent refreezing increased cohesion between snow grains. This totally stopped erosion of snow by wind. As the weather became cloudier, a thin layer of surface hoar or precipitated needle crystals deposited during the night (June 3 and 10, see Figure 4) were regularly observed at the beginning of the day before they rapidly melted. The underlying snow layers I and II, unaffected by surface melting, remained nearly unchanged (with SSAs of 10.6 ± 4.1 m$^2$ kg$^{-1}$ and 13.8 ± 6.9 m$^2$ kg$^{-1}$, and densities of 370 ± 26 kg m$^{-3}$ 418 ± 51 kg m$^{-3}$ for layers I and II, respectively). The SSA of layer III (24.7 ± 4.3 m$^2$ kg$^{-1}$) kept on decreasing during phase II (Figure 5) until it had completely transformed into wet grains (phase III). Solid melt-freeze layers, almost ice-like, (layer Vb, Figure 5) were first observed on May 29 and became more and more common, to the point that they were present everywhere at the end of phase II and





several of them could be found in the same snow pit. Actual ice layers, formed by percolating meltwater that froze at interfaces between layers because of the change in capillary forces, were also observed starting on Jun 6 (not shown in Figure 5). After June 4, freeboard became frequently negative, possibly due to the increasing mass of snow at the surface but also likely due to melting bottom ice caused by the warming sea.

*Phase III. Ripe snowpack.* At this stage, the snowpack was only composed of coarse rounded grains with the lowest SSA
values recorded ($4.6 \pm 1.2$ m$^2$ kg$^{-1}$). It was isothermal at 0°C and its thickness decreased rapidly. In 2016, contrary to 2015, a layer of liquid water up to 10 cm thick was found nearly everywhere at the base of the snowpack before snow melt-out. That layer was likely the result of the imbalance between the rapid input of snow melt water and the low drain capacity of sea ice rather than the result of a negative freeboard. The water was slightly salty to the taste, but much less than sea water. The salt may have come from the brine that was inevitably present at the ice surface when ice formed in the fall, and
wicked up the first snow layer (Alvarez-Aviles et al., 2008; Krnavek et al., 2012).

Table 1. Average values of SSA and density and corresponding standard deviations for each phase (SSA in m$^2$kg$^{-1}$ and density in kgm$^{-3}$). Layer numbers refer to those in Figure 6.

| | Phase I | | Phase II | | Phase III | |
|---|---|---|---|---|---|---|
| | SSA | Density | SSA | Density | SSA | Density |
| Layer II | $8.9 \pm 4.4$ | $372 \pm 51$ | $10.6 \pm 4.1$ | $370 \pm 26$ | | |
| Layer II | $12.1 \pm 1.8$ | $409 \pm 40$ | $13.8 \pm 6.9$ | $418 \pm 51$ | | |
| Layer III | $33.4 \pm 2.6$ | $276 \pm 38$ | $24.7 \pm 4.3$ | $340 \pm 49$ | | |
| Layer IVa | $49.3 \pm 5.9$ | $260 \pm 122$ | $36.3 \pm 18.7$ | | | |
| Layer IVb | | | $35.0 \pm 5.6$ | $214 \pm 14$ | | |
| Layer Va | | | $11.6 \pm 5.6$ | $346 \pm 37$ | $4.6 \pm 1.2$ | $406 \pm 15$ |

**3.3 Spectral Albedo**

All albedo spectra from the 2015 and 2016 field campaigns are summarized in Figure 7. They are displayed by phase in order to better illustrate their corresponding specific spectral signatures. Mean albedo values at 500 nm and 1000 nm are specified in Table 2 for each phase.

*Phase I. Cold, dry snow.* The highest albedos, $0.97 \pm 0.01$ at 500 nm, were measured above cold winter snow (Figure 3
and Figure 7). Values slightly decreased during this phase almost only in the infrared from 0.80 to 0.70 at 1000 nm. Spatial variability was low and the lowest albedo, 0.68 at 1000 nm, was recorded only above snow dunes, where fresh snow did not accumulate.

*Phase II. Surface melting.* Following the onset of wet snow metamorphism at the surface, albedo declined mostly in the infrared down to 0.45 at 1000 nm (Figure 7) while it remained high in the visible with all values>0.95 at 500 nm. Figure
7 shows large variations in albedo at 1000 nm. In particular, sudden increases are observed after snowfalls (May 30 and June 4 2015, for instance, Figure 3). These increases brought the albedo back near values observed in phase I. Despite the wider range of albedo in phase II compared to those in phase I (Figure 7), spatial variability did not increase during this phase because changes in snow SSA were homogeneous over the sea ice surface.





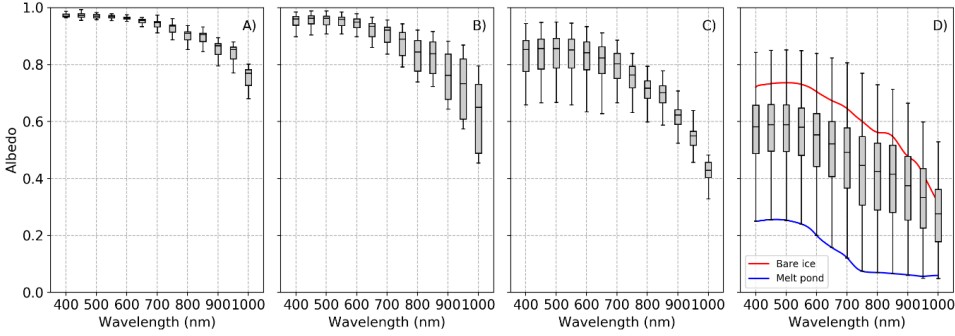

**Figure 7. Spectral albedo from 400 nm to 1000 nm for both 2015 and 2016 campaigns, represented with boxplot graphs and sorted by phase: A) cold, dry snow, B) surface melting, C) ripe snowpack and D) melt pond formation. In D) two specific albedo spectra over bare ice only and melt pond only are also shown.**

*Phase III. Ripe snowpack.* Darker patches and lower albedos were observed (Figure 2C) as snow thickness declined. Albedo decreased in the visible range, from 0.95 to 0.65 at 500 nm, while comparatively it remained steadier in the infrared, $0.43 \pm 0.042$ at 1000 nm (Figure 3). In Figures 3 and 7C, the albedo ranges in the visible are getting wider over time, showing that space variability appeared in this phase. This spatial variability in albedo followed spatial variability in snow thickness. Albedo in the visible was lower above thinner snowpacks.

*Phase IV. Melt pond formation.* The transition between phase III and IV was not sharp because melt ponds appeared suddenly while snow patches were still remaining. The albedo decreased over the whole spectral range (Figures 3 and 7D) down to 0.74 and 0.25 at 500 nm and 1000 nm, respectively, over bare ice, and 0.32 and 0.06 over melt ponds (Table 2). Spatial variability was maximal during this stage. The cooling event that brought less than one centimeter of snow on June 25 2016 temporarily enhanced albedo (Figure 3). This increase was largest in the near-infrared.

**Table 2. Mean albedos and corresponding standard deviations at 500 nm and 1000 nm during each phase. In phase IV, measurements above ice and pond only (one station for each) are specified.**

|  | Phase I | Phase II | Phase III | Phase IV | | |
|---|---|---|---|---|---|---|
|  |  |  |  | All | Ice | Pond |
| Albedo 500 nm | $0.97 \pm 0.01$ | $0.95 \pm 0.024$ | $0.84 \pm 0.073$ | $0.57 \pm 0.122$ | 0.74 | 0.32 |
| Albedo 1000 nm | $0.75 \pm 0.042$ | $0.63 \pm 0.121$ | $0.43 \pm 0.042$ | $0.27 \pm 0.120$ | 0.25 | 0.06 |

### 3.4 Radiative transfer modeling

#### 3.4.1 Snow impurity content

The particulate absorption spectra in melted snow samples collected during the field campaign were all very similar in terms of shape but differed in their magnitudes, which ranged from 0.03 m$^{-1}$ to 1.10 m$^{-1}$. The average spectrum derived





from the twelve samples is shown in Figure 8 as a thick grey line. It was multiplied by the average density of snow (350 kg m$^{-3}$) in order to obtain an average absorption coefficient of the impurities in the snow. The shape of this spectrum is

fairly similar to that of mineral dust (MD) (Caponi et al., 2017; Wagner et al., 2012), indicating that MD was the main LAP in snow at our site. The MD likely originated from the land around the ice camp, as granite cliffs are very common near Qikiqtarjuaq. The first simulations showed simulations could be improved by adding black carbon (BC) as an extra LAP. Refractive indices of the MD and BC, derived respectively from (Wagner et al., 2012) and (Bond and Bergstrom, 2006) were used in TARTES to simulates the LAPs. As shown in Figure 8, a mixture (red dashed line) of 1700 ng g$^{-1}$ of

MD and 14.4 ng g$^{-1}$ of BC correctly simulates, in term of shape and amplitude, the average particulate absorption measured within the snow.

As particulate absorption of snow was only measured during the 2016 campaign, we used our 2015 vertical irradiance profile obtained in five snow dunes in order to investigate the presence of such impurities within the snow in 2015. LAP concentrations were obtained from the exponential rate of irradiance decrease in a homogeneous layer following (Tuzet et

al., 2019) and (Belke-Brea et al., 2021). Making our calculation on the most linear parts of the five logarithmic profiles we found an average concentration of 886 ng g$^{-1}$ of MD and 14.1 ng g$^{-1}$ of BC in the top layer of the dunes. Those concentrations are within a factor of two (for MD) or similar (for BC) to those derived from the LAP absorption study the following year.

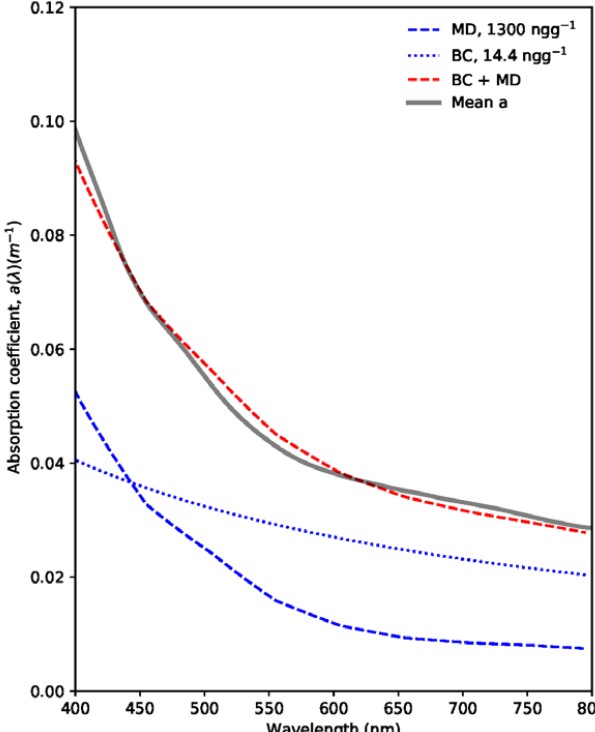

**Figure 8. Absorption coefficients of snow impurities in m$^{-1}$, measured in melted snow. The thick grey line denotes the average of all particulate absorption measurements for snow with a density of 350 kg m$^{-3}$. Blues lines denote the coefficients of MD (dashed), BC (dotted), and the sum of both (dashed red), used to match the measurements, with 1700 ng g$^{-1}$ of mineral dust and 14.4 ng g$^{-1}$ of black carbon**





### 3.4.2 Albedo Modeling

Albedo simulations were performed to assess the adequacy of the snow observations collected for radiative transfer modeling, and to quantify the sensitivity of surface albedo to snow surface properties and snow depth. Albedo was simulated for each snowpit using vertical profiles of SSA and density (Figure 6) as inputs in the TARTES model. A single impurity concentration was used for all albedo calculations because we had no way to determine specific concentrations in each snowpack. It is possible to fit each spectral albedo with optimized LAP concentrations, but without measurement

in each pit, this is not very meaningful. We used a mixture of 2000 ng g$^{-1}$ of MD and 20 ng g$^{-1}$ of BC for all snowpits. These values were chosen to optimize the albedo fit at 500 nm during phase I, where snow physical measurements are the most reliable. Increments of 500 ng g$^{-1}$ were tested for MD. The 100:1 ratio between MD and BC was chosen for consistency with measurements from irradiance profile in snow dunes (ratio 118:1) and data from melted snow (ratio 63:1). Impurity concentrations are not necessarily identical in dunes and in the rest of the snowpack for many reasons, one of

which being that layers in dunes did not necessarily deposit at the same time as layers with similar grain shapes elsewhere. Furthermore, LAPs and in particular MD particles were probably for the most part deposited during wind events and they may have been diluted in the thicker dune layers relative to elsewhere. Lastly, LAP concentrations in 2016 may have been slightly different from those in 2015. The values used, although slightly higher than measurements, are thus reasonable.

Only data from the 2015 campaign are presented since this dataset is more comprehensive and covers the first three phases.

As mentioned in the previous section, snow grain size impacts albedo mostly in the near infrared while the impact of snow thickness is mainly observed in the visible. Thus, both measurements and simulation results are compared at 500 nm and 1000 nm. They are presented in Table 3 and Figure 9 where simulations with and without LAPs are shown.

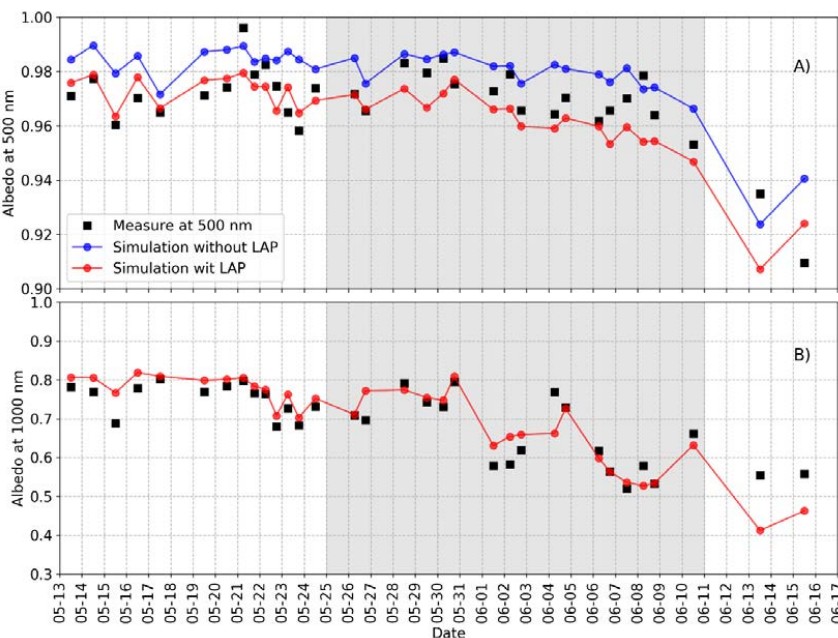

**Figure 9. Albedo measurements (black squares) and simulations (red and blue dots) at 500 nm (A) and 1000 nm**
**(B) for each sampling station in 2015 (different scale in y axis). Blue dots are simulations without impurities. Red dots include impurities (mineral dust and black carbon). At 500 nm, impurities decrease simulated albedo by 1 to 2% whereas at 1000 nm their effect is hardly detectable, so that the red curve conceals the blue one. The gray shaded areas indicate the surface melting period.**





The largest differences are for the last two dates. These are during phase III when melting was extensive and rapidly
evolving, with visible short-scale variations. Since albedo measurements cover about 1 m$^2$ while density and SSA values
are from point measurements, an imperfect coincidence between both data sets is expected. Albedo and snowpit
measurements are also done in sequence, so that variations between both measurements are possible, perhaps even likely.
If we remove these last two measurements from the comparison, Figure 9 shows that at 500 nm, the difference between
measurement and model is <1% (absolute value) for 25 out of 31 values. At 1000 nm, the difference is <0.04 (absolute
value) also for 25 out of 31 values, and <0.02 for 16 values. During phase I, average simulations at 500 nm are essentially
identical to measurements, while during phase II they are below measurements by 0.8%. At 1000 nm, these values are
3.6% and 0.9%.

**Table 3. Relative deviation between albedo simulations and measurements at 500 and 1000 nm, in percentages and corresponding standard deviation.**

|  | All results | Phase I | Phase II | Phase III |
|---|---|---|---|---|
| Albedo at 500 nm | 1.1 ± 0.9 | 1.2 ± 0.9 | 0.9 ± 0.6 | 1.1 ± 2.3 |
| Albedo at 1000 nm | 0.7 ± 7.6 | 3.7 ± 2.5 | 0.9± 6.4 | -21.3 ± 4.2 |
| Albedo at 500 nm, with LAP | -0.5 ± 1.0 | 0.0 ± 0.7 | -0.8 ± 0.6 | -0.7 ± 2.3 |
| Albedo at 1000 nm, with LAP | 0.7 ±7.6 | 3.6 ± 2.5 | 0.9± 6.4 | -21.3 ± 4.2 |

Given these reasonably good fits, simulations were used to calculate three metrics relevant to radiation and energy of the
(snow + sea ice) system: the broadband albedo, the broadband radiative energy input into the snow, and the flux of visible
radiation at the snow-ice interface. Calculations were performed for two wavelength ranges: the solar spectrum (300-3000
nm, i.e. broadband) and photosynthetically active radiations (PAR, 400-700 nm). This last metric is relevant to
understanding phytoplankton blooms under sea ice (Ardyna et al., 2020). Time series of these metrics are plotted in Figure
10. Usual units for PAR are photons fluxes and these are shown in Figure 11. Units of W m$^{-2}$ are used in Figure 10 for
consistencies and to allow easy comparisons.

The broadband albedo reached a maximum of 0.87 during phase I (dry snow). During phase II (surface melting), its
minimum value was 0.76 near the end of the period, inducing an increase in the solar input from 103 W m$^{-2}$ to 185 W m$^{-2}$
under an incident clear-sky flux of 784 W m$^{-2}$, as detailed in section 2.6. The succession of snowfalls and melting episodes
caused significant variations in solar radiation transmitted to the system. The layers of fresh snow reduced the energy input
by approximately 20 W m$^{-2}$ on May 30, 16 W m$^{-2}$ on June 4 and 11 W m$^{-2}$ on June 10. On the other hand, the rapid
metamorphism of the snow resulted in an increase in solar input of 50 W m$^{-2}$ from May 30 to June 1 and 45 W m$^{-2}$ from
June 4 to 6. Regarding PAR, the flux at the snow-ice interface remains most of the time below 2 W m$^{-2}$ during phases I
and II. This is because essentially all visible radiation is reflected. Figure 10C indeed shows that the PAR flux at the snow
surface is <10% of the broadband flux. Only when the snowpack thickness decreases at the end of phase II can significant
visible radiation reach the ice surface and be transmitted downwards. The PAR flux on 13 June reaches 22 W m$^{-2}$.

PAR fluxes are used to evaluate biological productivity and considering them is especially important in this case to relate
their increase to the initiation of the phytoplankton bloom in the water column. Figure 11 plots PAR fluxes at the snow-
ice interface simulated in this study together with underwater PAR fluxes at 1.3 m depth and water column-integrated
chlorophyll *a* amounts, reported by (Massicotte et al., 2020).

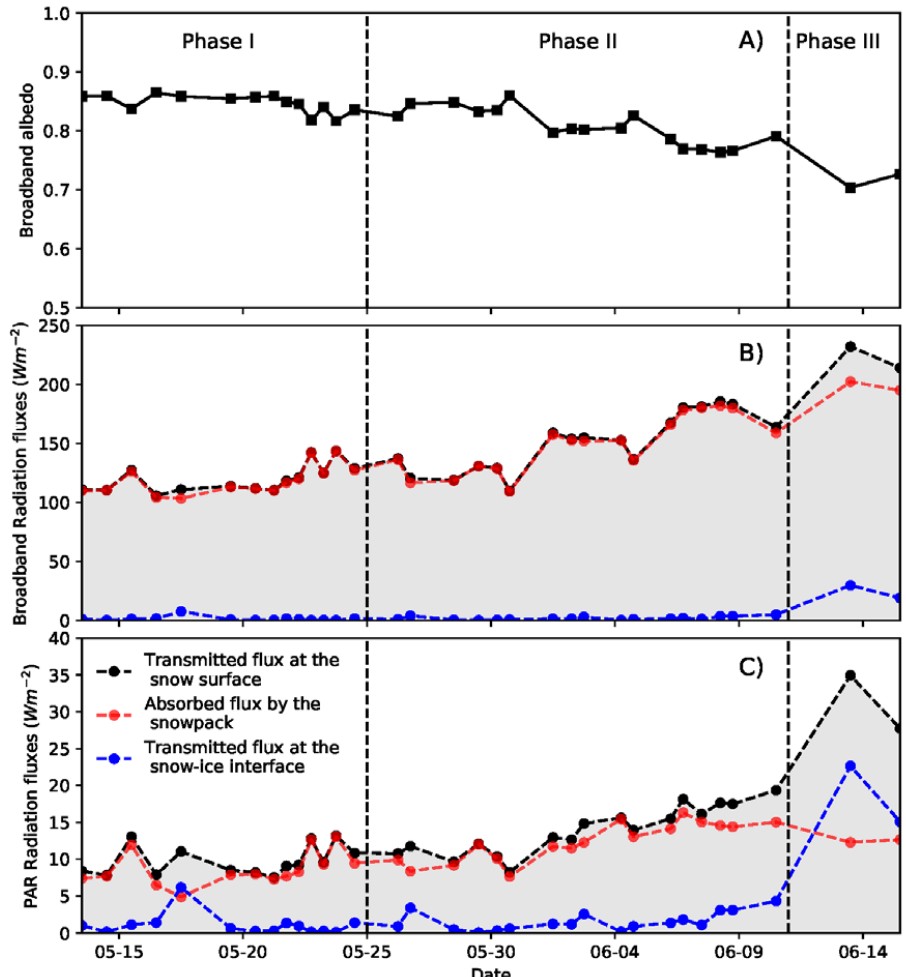

**Figure 10. Simulated radiative metrics of the (snow + sea ice) system. A) Broadband albedo (300-3000 nm) of the system. B) Broadband radiative fluxes, based on downwelling radiation calculated on 1 June 2015 using the SBDART model. C) Photosynthetically active radiation (PAR: 400 – 700 nm) flux at the snow-ice interface.**

## 4 Discussion

The previous section detailed the evolution of physical properties and albedo of snow. It also tested our ability to retrieve albedo by radiative transfer modeling using measured snow properties. In this section we first propose to reconstruct the main steps of formation of the snowpacks observed during samplings. We then discuss the co-evolution of albedo and of snowpack characteristics. Finally, we discuss possible future improvements in the accuracy of albedo simulations.

### 4.1 Snowpack formation

Arctic snowpacks, whether on land or sea ice, almost always show the same type of structure, with basal layers of depth hoar and sometimes faceted crystals, and top wind slabs with occasional fresh or recent snow that usually gets wind-blown





to add more sub-layers to the top wind slab (Derksen et al., 2009; Domine et al., 2016b; Domine et al., 2012; Sturm et al., 2002). Depth hoar forms early in the season when the elevated temperature gradient between the warm soil or young thin sea ice and the cold air is large (up to 200 K m$^{-1}$) and generates upward water vapor fluxes that allow the growth of large depth hoar crystals. Depth hoar can be soft or indurated, depending on the initial density of the snow when it started forming (Domine et al., 2016b). On the North American-Greenland side of the Arctic, there is usually more precipitation

and snow accumulation in fall and spring, whereas winter is rather dry (Domine et al., 2021; Sturm et al., 2002; Warren et al., 1999).  The depth hoar layers (Layer I, Figure 5) thus most likely formed in fall. They were indurated and of high density, indicating that they formed from dense wind slabs. The layers of faceted crystals (Layer II) formed subsequently also in the fall, when the temperature gradient had decreased because the snowpack was thicker and the surface of the sea ice was colder because it was thicker. The fact that they formed in dense wind slabs probably explains why they did not

reach the depth hoar stage, as crystal growth under temperature gradient metamorphism is slower at higher snow densities (Flanner and Zender, 2006; Marbouty, 1980). The density of the lower layers indicates windy meteorology in the fall, with winds frequently exceeding 10 m s$^{-1}$. (https://weatherspark.com/h/y/147443/2015/Historical-Weather-during-2015-at-Qikiqtarjuaq-Airport-Canada#Figures-Summary, accessed on 7 March 2022)The windy character of the area in the fall is confirmed by the presence of whale-back dunes (Filhol and Sturm, 2015) which probably formed mostly in the fall, given

the predominance of depth hoar and faceted crystals in their stratigraphy.

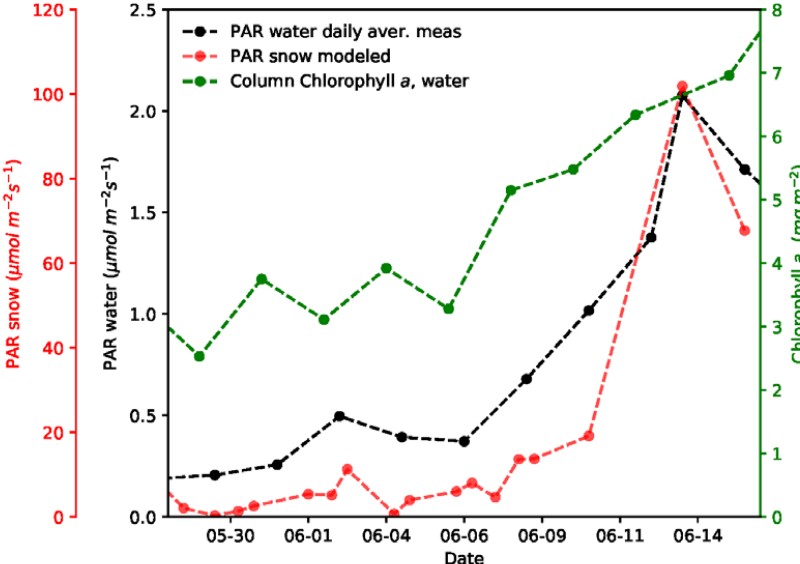

**Figure 11. Simulated PAR fluxes at the snow-ice interface, compared with PAR fluxes under the sea ice at 1.3 m depth and with chlorophyll *a* concentrations integrated over the water column. These last two data sets are from**
**Massicotte et al. (2020).**

Wind slabs (Layer III) formed when the temperature gradient had decreased too much to allow the formation of depth hoar or even faceted crystals. Wind slabs may have formed on whale backs as early as late fall, since the snow thickness reduced the temperature gradient. However, Figure 4 shows significant accumulation in spring, and this is probably when most of the wind slabs formed. This is confirmed by their fairly high SSA, often >30 m$^2$ kg$^{-1}$. This is higher than for typical wind



slabs (Domine et al., 2016b; Domine et al., 2012), suggesting the SSA did not have time to decrease after formation.

Depth hoar almost always has a SSA close to 10 m$^2$ kg$^{-1}$, while the SSA of faceted crystals is not much higher, often <15 m$^2$ kg$^{-1}$ (Domine et al., 2016a; Domine et al., 2002; Taillandier et al., 2006) as observed here (Figure 6). Wind slabs have a higher SSA, usually in the range 25-35 m$^2$ kg$^{-1}$ in most places, and this enhanced albedo in the infrared (Domine et al., 2006; Wiscombe and Warren, 1980). However, we note with interest that the lowest SSA values for wind slabs are found

on whale backs, with values often <25 m$^2$ kg$^{-1}$. Since SSA of rounded grains decreases over time (Taillandier et al., 2007), this tends to confirm that the wind slabs of whale backs formed earlier than elsewhere, and probably in late fall as suggested above.

In summary, most of the whale back snow height accumulated in fall under strong winds and initially elevated temperature gradients, producing depth hoar and faceted crystals. A significant part of the general snowpack also formed in fall and

the strong temperature gradients also produced depth hoar and faceted crystals there. Wind slabs probably started forming in fall but the high SSA in most spots indicate recent formation, in spring, just before or during our campaigns (Figure 4). Based on general western Arctic climatology and on snow depth data of Figure 4, we propose that snow accumulation in winter was minimal.

### 4.2 Albedo and surface evolution

Our data illustrate the co-evolution of albedo and of the state of the snowpack. Albedo was initially high as long as the snowpack remained dry. It gradually decreased after melt onset as surface snow grains coarsened and this decrease continued until the snowpack completely vanished. During our campaigns, albedo was affected by synoptic weather, such as rain and snowfall events. Snowfalls were common during the melting period. They suddenly and temporarily increased the albedo, substantially delaying melt-out.

The data shown in Figure 7 and Figure 9 show a different evolution of spectral albedo during each phase. Phase I sees little change in albedo, either in the visible or infrared ranges because the snowpack remains dry. Its thickness does not change and the small drop in snow SSA (Figure 6) just leads to a modest drop in the infrared albedo. Phases II and III show strikingly different evolutions. In phase II, the albedo decrease is mostly in the infrared. In this wavelength range, albedo is mostly sensitive to grain size in surface layers (Domine et al., 2006; Wiscombe and Warren, 1980), and the

melting of these layers results in the decrease in SSA (i.e. increase in grain size), as shown in Figure 6, which reduces the infrared albedo. In the visible, albedo is much less sensitive to grain size. The physical variable most likely to affect its value is the snowpack thickness. (Wiscombe and Warren, 1980) (their Figure 13) show using calculations that for large snow grains (1 mm, fairly similar to melt grains we observed) the albedo at 500 nm decreases in a detectable manner when the snowpack water equivalent decreases to 100 mm. This corresponds to a depth of about 28 cm at our site, a value

commonly observed (Figure 6). Thus, snow depth during phase II mostly remains above the threshold where visible albedo would be affected.

In phase III on the other hand, the infrared albedo varies little because SSA has essentially reached its terminal value of a few m$^2$ kg$^{-1}$. In the visible however, the albedo is affected by the decrease in snowpack thickness. Radiation reaches the sea ice where it is more absorbed than on snow, and the visible albedo decreases as the snowpack thins down. In phase IV,

the albedo decreases in both the infrared and the visible, because snow is replaced by ice and melt ponds. The trends in phase II and III were perturbed by snowfalls, but overall the dynamics of the spectral albedo appears to be a good indication of the phase of the snowpack evolution on sea ice.



The results concerning the overall evolution of the broadband albedo (Figure 10) is in line with previous observations (Gallet et al., 2017; Grenfell and Perovich, 2004; Nicolaus et al., 2010; Perovich et al., 2002). However, our detailed
measurements of snow physical properties allow a more detailed interpretation of spectral albedo evolution, in particular effects related to the decrease in SSA.

These results suggest that impurities can be important on the fast ice to predict melt dynamics and transmission through the column. Because LAPs tend to concentrate during melt and because of the coupling with the grain size (Kokhanovsky, 2013), shortwave absorption can be greatly enhanced, further amplifying melt. LAPs also have a peculiar impact on the
PAR available under the sea-ice, as very small amount of impurities are sufficient to reduced by orders of magnitudes the transmittance in the blue-green region (Tuzet et al., 2019). While this effect may concern a small fraction of the Arctic sea-ice cover, it seems important enough to be further investigated.

**4.3 Albedo modeling, current limitations**

Figure 9 shows an overall very good ability of our model to simulate measured spectral albedo. Snow density and SSA
used in modeling were measured values and were not adjusted. For simplicity, single values for impurity concentrations (one for MD and one for BC) were used for the whole field campaign. Although these values were optimized, they reflect values measured in whalebacks in 2015 and in melted snow samples in 2016 within a factor of two. This is acceptable given the uncertainties related to LAPs in general (Tuzet et al., 2019). Further improvements in modeling could be obtained by optimizing B and g. We used B and g values for spheres (Libois et al., 2013) because most of the time surface snow
crystals were rounded by melting. Today, however, we have little understanding of how grain shapes affect B and g (Libois et al., 2014). Ideally, different values would be used for each snow layer. This clearly has the potential for improving the agreement between simulations and measurements, especially in the infrared. Attempts using different B and g values using TARTES indeed show that improvements are possible. However, since we currently would have little justification for selecting such values, we feel that this purely fitting exercise would be pointless. At present, our simulations are most
of the time within 1% of measurements in the visible and within 2% in the infrared. Improving this agreement could be done by performing chemical measurements of impurities in each snow layer and by improving our understanding of the relationship between grain shape and the physical variables B and g. This latter suggestion could be realized either by performing field studies similar to (Libois et al., 2014) or ray-tracing simulations on microtomography images of real snow samples (Haussener et al., 2012; Letcher et al., 2021).

**4.4 Radiative fluxes and phytoplankton blooms**

Figure 11 shows that the PAR flux increase at the snow-ice interface around 9 June coincides fairly well with the chlorophyll *a* increase. (Massicotte et al., 2020) report time series of average daily PAR values in the sea water at 1.3 m depth (also shown in Figure 11) and indicate that the initiation of the phytoplankton bloom was around 8 June. The PAR under-ice increase curve is somewhat different from our modeled curve, but this can probably be explained by different
snow-melt phenology at our snow measurement sites, which are different for each day, and at the PAR measurement sites. The coincidence between the increase snow transmittance and the increase in chlorophyll *a* is nevertheless pretty good.

**5 Conclusion**

Snow over sea ice was intensively studied during two melt seasons in Baffin Bay. These studies include spectral albedo measurements and vertical profiles of physical properties of snow that are relevant to radiative transfer modeling: density



and SSA. The entire transition from cold and dry snow covers in early spring to ponded sea ice in late spring was recorded.

Both years, albedo evolved following four main phases related to the conditions of the snow cover. During these phases, broadband albedo was first high, up to 0.87, over a dry snowpack (phase I) composed of basal layers of indurated depth hoar and faceted crystals topped by one or several wind slabs with sometimes a layer of fresh snow. Albedo gradually decreased in the near infrared as snow grains coarsened because of surface melting (phase II). At some point, the snowpack

became ripe and isothermal, its thickness decreased faster leading to a decrease in albedo in the visible range for the first time (phase III). This drop in albedo was due to the influence of the underlying darker sea ice as the light penetration depth in snow increased and snow depth decreased. Spatial variability appeared and was directly linked to the snow thickness and the optical properties of the underlying media. Melt ponds formed with snow melt-out during phase IV.

Snow physical properties and impurity contents were used as inputs to a radiative transfer model in order to simulate the

albedo. Simulations reproduced the measured time series of albedo very well and demonstrated the interest of measuring time series of snow SSA vertical profiles to understand albedo evolution. Other important factors identified here are the impurity content and snow depth, which have a strong impact in the visible range, and the geometric factors B and g, which are today poorly constrained for real snow.

Lastly, this work allowed the calculation of the spectral radiative fluxes at the snow-ice interface. In particular, calculations

showed that PAR fluxes at this interface rose durably above 9 µmol m$^{-2}$ s$^{-1}$ only at the end of phase 2 (11 June), when the snowpack thickness started decreasing. The timing of this PAR increase coincided fairly well with the initiation of the phytoplankton bloom, as seen in Figure 11, and to the date mentioned by (Massicotte et al., 2020), also 11 June in their Figure 5. This suggests that modeling radiative transfer through snow, based on accurate data on snow physical and chemical properties, may allow the reliable prediction of the timing of phytoplankton blooms.

**Acknowledgements**

The field campaign was successful thanks to the contribution of J. Ferland, G. Bécu, C. Marec, J. Lagunas, F. Bruyant, J. Larivière, E. Rehm, S. Lambert-Girard, C. Aubry, C. Lalande, A. LeBaron, C. Marty, J. Sansoulet, D. Christiansen-Stowe, A. Wells, M. Benoît-Gagné, E. Devred and M.-H. Forget from the Takuvik laboratory, C.J. Mundy and V. Galindo from University of Manitoba & F. Pinczon du Sel and E. Brossier from Vagabond. We also thank Michel Gosselin, Québec-

Océan, the CCGS Amundsen and the Polar Continental Shelf Program for their in-kind contribution in terms of polar logistics and scientific equipment.

**Funding information**

The GreenEdge project was funded by the following French and Canadian programs and agencies: ANR (Contract #111112), ArcticNet, CERC on Remote sensing of Canada's new Arctic frontier, CNES (project #131425), French Arctic

Initiative, Fondation Total, CSA, LEFE and IPEV (project #1164). This project would not have been possible without the support of the Hamlet of Qikiqtarjuaq and the members of the community as well as the Inuksuit School and its Principal, Jacqueline Arsenault. The project is conducted under the scientific coordination of the Canada Excellence Research Chair on Remote sensing of Canada's new Arctic frontier and the CNRS & Université Laval Takuvik Joint International Laboratory (UMI3376).

**Competing interest**



Florent Domine is a member of the editorial board of The Cryosphere.

**Authors contributions**

MB, GP and FD designed research. MB obtained funding. GV performed research with assistance from GP and FD. LA and GP designed and built the Solalb instrument. GV and FD wrote the paper, with inputs from GP, MB and LA.

**Data and code availability**

The data used here are available at https://doi.org/10.17882/59892 (Massicotte et al., 2019). The TARTES model is available from https://github.com/ghislainp/tartes.

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
