# Peer review of "Metamorphism of snow on Arctic sea ice during the melt season. Impact on spectral albedo and radiative fluxes through snow"

_The Cryosphere, 2022_

## Author Comment (AC1)

Reply to Anonymous Referee #1

*We copied the review below and imbedded our responses within.*

**General comments**

The study by Vérin and colleagues investigates snow metamorphism on the arctic sea ice and in particular how it impacts on spectral albedo, and how well the albedo variability can be reproduced using a RTM. I believe that this study provides great insights into the temporal variability of snow sea ice albedo during the melt season, and that the albedo dataset could be useful to parametrize or validate the evolution of snow sea ice albedo in global models. The results are clearly presented and the methods are robust - my comments are minor and mostly focused on the albedo measuring and modelling since snow metamorphism is out of my expertise.

*We thank the Reviewer for this positive appreciation and for constructive comments.*

**Specific comments and technical corrections**

Title: Arctic marine snow can be confusing because it commonly refers to debris sinking in the ocean, maybe it would be clearer to indicate "Arctic snow on sea ice" or equivalent in the title?

*We have changed this phrase to "snow on Arctic sea ice" in the title*

**Abstract:**

Line 19-20: Maybe a minimum BBA value can be indicated here rather than the value at 1000nm? Then it makes it comparable with the number from line 18.

*Indeed. We have replaced this with the BB value (0.76, line 20).*

Line 24-25: "based on measurements" is probably unnecessary here because "measured" is already written earlier in the sentence - maybe reformulate to ".. using measured SSA, density vertical profile and impurity content"?

*Actually, impurities were not measured for each profile unlike SSA and density. We changed the wording to "Spectral albedo was simulated by radiative transfer using measured SSA and density vertical profiles, and estimated impurity contents based on limited measurements.", line 24.*

Line 25: "calculationS" - also, "at the interface snow-ice" may be clearer than "at the top of the sea ice"?

*Thank you. Changed.*

**Introduction:**

Line 56-57: What about longwave radiation on overcast days – can this also have an impact on the dry snowpack?

*The important variable is temperature, which is determined by the energy budget. LW radiation is one term of this energy budget, and special consideration of this term is not necessary here.*

Line 70: In Domine, F., et al. "Three examples where the specific surface area of snow increased over time." *The Cryosphere* 3.1 (2009): 31-39, the authors conclude that "SSA increases are probably not rare", including cases that are described in this paragraph like depth hoar formation. Is this reference maybe worth mentioning here? (especially as other "occasional" processes are mentioned in the introduction, such as summer snowfalls in line 86.

*Sure, SSA increases are not rare but the general and most common process is SSA decrease. In Domine 2009, depth hoar formation leads to SSA increase only when it forms from melt-freeze crusts, which is not the case here. Summer snowfalls add new snow, so this cannot be considered as the SSA increase of a given snow layer. Therefore, citing Domine (2009) does not appear justified here. (Domine is writing this).*

Line 74: "samplings" is probably incorrect, maybe "a lot of samples" or "a lot of sampling"?

*"samplings" seems fine to us.*

Line 93: global radiative transfer "in" sea ice not "of"?

*Sure, changed to "in". Thank you.*

Line 101: "light absorption" may be clearer than "optical absorption"?

*Sure, changed to "light". Thank you.*

Line 102: instead of "and using measurements on melted snow samples", maybe "and measuring absorption coefficients of LAPs isolated from snow samples"?

*Thank you, suggestion followed (line 103).*

Line 116: I think that the correct spelling is "set up", not "setup"?

*Setup is in the dictionary, with the intended meaning.*

Line 157: LAPs are often distributed in the top cms – if the top 7cm are discarded, how is it possible to get data on the LAPs from the profiles? Or do the authors mean that the top 7cm was removed only from thin snowpacks on line 150, which is why thick snowpacks were used for retrieval of LAPs properties?

*LAPS are preferentially distributed in the top cm mostly in melting snow. In dry snowpack, deep dark layers are very common, see e.g. (Kaspari et al., 2014). We do not say that the top 7 cm were discarded, only that "profiles were normalized to the irradiance at a depth of 7 cm". In any case, irradiance profiles were used to determine the type of impurities based on their spectra, rather than the actual absorption coefficients, which were determined from impurities extracted from melted snow. The new wording lines 160-163 reflects this. "Analysis of the profiles yield the spectral absorbance of LAPs, which were used to determine the type of absorbing particles (essentially black carbon (BC) or mineral dust). LAPs were also analyzed using chemical methods to obtain absorption coefficients, as detailed in the following section."*

Line 180: Is it possible to indicate the reference of the electronic scale and the sensitivity?

*We specify that the resolution of the scale was 0.1 g (line 185). The reference of the scale is not known at present.*

Line 196: "then" instead of "them"

*Changed, thank you.*

Line 207: "As the snowpack was already ripe, the study of spatial variability using large scale measurements was favored." What do large scale measurements mean here – UAV measurements, or transect measurements?

*Changed to "transect measurements" line 212.*

Line 213: short wavelengthS

*Changed, thank you.*

Line 223: Is it possible to indicate at what time were the albedo measurements recorded?

*Albedos were recorded at various times during the day. We unfortunately cannot specify the time of each measurement. SZA was between 47 and 57°, as specified line 244.*

Line 238-240: Why were simulations performed considering diffuse radiation and not using the exact SZA values corresponding to each albedo measurement?

*As stated line 243, diffuse radiation is equivalent to direct light with SZA ~ 50°. As stated line 244, the SZAs of our measurements were always between 47° and 57°, close to 50°. We therefore make the implicit approximation that SZAs between 47 and 57° are equivalent to 50°. Sky conditions were not always fully overcast of bright blue. A variable cloud cover was often present. Using the actual SZA would therefore not be necessarily better than our approximation. To clarify this, we write line 244 "simulations were performed by approximating the illumination conditions with diffuse radiation." Note alos that the following sentence reads "Doing so, the maximal error on albedo is ~0.01 at 1000 nm." To further justify the approximation.*

Line 252: "whose" to replace with "of which"?

*Merriam Webster grammar specifies "animals and objects, lack a possessive form, so* whose *can be used for their possessive forms as well, as in "the movie, whose name I can't remember." Please see* https://www.merriam-webster.com/words-at-play/whose-used-for-inanimate-objects' . *"whose" is therefore correct here and we keep this as is.*

Line 259-260: So density, SSA and irradiance profiles were not measured in 2016? Please indicate this in the methods when describing the sampling and analysis of snow physical properties (2.3 and 2.2.2)

*Line 149, section 2.2.2, we added "in 2015". Line 180, section 2.3, we added "These were performed in 2015 and not in the melting 2016 snowpacks."*

Figure 3: How were the wavelengths of 500 and 1000 for albedo chosen? Why not calculate broadband albedo in the IR and VIS? Or an averaged value?

*Reviewer 2 has made a related comment so we need to address this in detail.*

*First, why use 500 and 1000 nm?*

*The reason for presenting the results for two wavelengths at 500 and 1000 nm is two-fold:*

*- a single wavelength is a true surface property, independent of the incident solar spectrum (especially cloudy vs clear-sky conditions) and is easier to compare or to generalize than wideband albedo (integrated in the VIS or NIR) which depends on illumination conditions.*

*- 500nm and 1000nm are round numbers representative of the VIS and NIR domains. They are respectively close to the minimum of absorption (450-550nm) where SSA has a minimum effect and the first main absorption feature of the ice (1030nm) where SSA has a predominant effect. They are also in the domain of sensitivity of our spectrometer.*

*Of course, an infinite number of wavelength combinations can be investigated: VIS, IR, PAR, MODIS bands, bands from other instruments on various satellites, etc. These wavelengths combinations would require additional Figures that would lengthen an already long paper. More importantly, it would detract the reader's attention from our actual focus: investigating the relationship between snow optical and physical properties.*

*In conclusion, for our purpose, linking physical and optical properties, we believe the wavelength chosen are sensible although probably not unique, but choices have to be made. To keep pour paper focused and concise, we much prefer to limit our results presentation and discussion to these 2 wavelengths. Note however that we do discuss the full solar spectrum and PAR in Figures 10 and 11, and we believe this extension is sufficient. Numerous other Figures could be added, but we really do not wish to lengthen our paper and we need to stay focused on our purpose.*

Is it possible to indicate more clearly the different phases in the figure - eg the end of phase 3 in 2015 seems to extend to end of June on the figure but extends to mid-June in the text?

*2015: we left on 18 June, so we have no data beyond that date. We have now specified this in the Figure caption (line 271) to make this clear.*

Similarly, phase II starts on May 19 in the text but before may 14[th] in the figure.

*Phase 2 starts on May 25 in the text (line 286), and in the Figure as well.*

The albedo of the highly heterogeneous ponded sea ice from 2016 was calculated from the transect measurements?

*It was calculated from all available measurements in 2016 and these were mostly transects. We do not think it is critical to detail this.*

Is it possible to indicate in the figure legend how many measurements are included in the box plots?

*This was already done in the lower panel of Figure 2, which specifies the number of samples corresponding to each data set.*

Line 294: Did you mean "The transition from snow cover to bare ice"?

*Yes, thank you. Changed line 299.*

Line 356: Typo "junE 6"

*Changed, thank you. Line 364.*

Line 365: "wicked up the first snow layer" I don't understand the meaning of this, could it be reformulated?

*We added "by capillary rise" line 372.*

Table 1: Is it possible to indicate in the methods or in the table how many samples were analysed for density and SSA to derive averages and standard deviations?

*The number of samples has been added in parentheses in Table 1.*

Figure 7: Why are albedo data presented only from 400nm if the spectroradiometer could measure from 300? Is it possible to indicate the number of samples used in the boxplots, or at least a range? What does "specific albedo spectra" mean, are they averages, or how were these examples of bare ice and melt pond albedos chosen?

*The signal was too low below 400 nm. We changed 300 to 400 line 128. The number of samples has been added to the Figure 7 Caption. We changed "two specific albedo spectra over bare ice only and melt pond only are also shown." to "one albedo spectrum over bare ice and one spectrum over a melt pond are also shown." in the Figure 7 Caption.*

Line 409: I am not sure I fully understand what was done here: "It was multiplied by the average density of snow (350 kg m -3 ) in order to obtain an average absorption coefficient of the impurities in the snow". If I understand correctly, the absorption of particulates from melted snow leads to a coefficient expressed in m-1 of melted snow.

How can this then be multiplied by the density of snow, leading to units of kg snow per m4 of snow, and give an "average absorption coefficient"? and where are these "average coefficients" shown? Do the authors mean that they divided the coefficient in m-1 of melted snow by the density of water and then multiplied it by the density of the snowpack in order to get an absorption coefficient in m-1 of snow instead of melted snow?

If possible, would it be possible to divide the absorption coefficient in m-1 by the LAP concentration (in kg m-3) in the solution of melted snow that was filtered to carry the spectrophotometric analysis in order to get a mass absorption cross section in m2 kg-1? Then the data could be used in other widely used radiative transfer model such as SNICAR.

*We apologize for the lack of clarity. We now specify line 421 that "It was divided by the density of water (1000 kg m$^{-3}$) and multiplied by the average density of snow (350 kg m$^{-3}$)". Units therefore remain in m$^{-1}$.*

Line 434: "It is possible to fit each spectral albedo with optimized LAP concentrations, but without measurement in each pit, this is not very meaningful" I do not understand what the authors mean here – if it is possible to retrieve the impurity concentrations by inversing TARTES, why would it not be meaningful?

*Indeed, it is always possible to invert TARTES to retrieve a hypothetical impurity concentration for each profile. However, our preferred approach, mentioned line 428, has*

*been to use a single value for all snowpits, close to the limited number of measured values. This is certainly debatable, but no more than fitting each profile without measurement of impurities in each profile.*

Line 438: "consistence" -> "consistency"

*Thank you. Changed, line 455*

Line 444-447: It would be clearer to have this paragraph at the beginning of the section 3.4.2 to understand the reasoning in comparing LAPs concentrations between 2015 and 2016 in line 443.

*We agree. This paragraph has been moved to the start of section 3.4.2. Thank you.*

Figure 9: typo "albedo witH LAP" (h missing) and maybe write "LAPs" instead of "LAP" because both MD and BC are included in the simulations if I understood correctly. It would be beneficial to add the phases as in Figure 3.

*The legend box has been corrected as requested. Phases are indicated by grey shading, as explained in the caption.*

Lines 458-462: It would be clearer to indicate only absolute values without units in this paragraph since the relative errors in % are in the table already – the error at 500nm is given in % (indicated as absolute value?) whilst the 0.04 and 0.02 are indicated without % units (are they %?).

*We apologize for this unclear writing. All values are absolute, there should not be any %. This has been modified. Lines 477-480.*

Line 467: Why were LAPs omitted in these simulations (Figure 10) if they improve the fit between measured and modelled albedo?

*LAPs were not omitted.*

Line 512-513: the link in parenthesis should come before the dot and there should be a dot and a space after the last parenthesis

*Changed, thank you. Line 531*

557-558: According to figure 3, the albedo at 1000nm still varies a lot in phase III (similar slope than phase II?)?

*It seems that the Reviewer is mixing up data from 2016 and 2015. The dates for Phase 3 are different in both years. In Figure 3, if one considers only data for Phase III in 2015 and in 2016, paying attention to the different dates, it is clear that the albedo at 1000 nm for Phase III varies little. Of course. If the dates for phase III in 2015 are used to analyze albedo data for 2016, it does appear that albedo varies a lot, but this is clearly an error. The reduce the risk of error we have reworded line 576 as ", as visible in Figure 3 for both 2015 and 2016 and in Figure 7c."*

Line 568: What is meant by "the coupling with the grain size"?

*We changed "coupling" to "enhanced impact of LAPs for low SSA values". Line 588.*

Line 575: What is meant by "not adjusted"?

*We changed to "used without adjustment". We feel that the term "adjusting" to described fitting values in models is pretty common. Line 595.*

Line 600: snow cover instead of coverS

*We believe our writing is correct. Line 620.*

Reference

Kaspari, S., Painter, T. H., Gysel, M., Skiles, S. M., and Schwikowski, M.: Seasonal and elevational variations of black carbon and dust in snow and ice in the Solu-Khumbu, Nepal and estimated radiative forcings, Atmos. Chem. Phys., 14, 8089-8103, 2014.

---

## Author Comment (AC2)

Reply to Anonymous Referee #2

*We copied the review below and imbedded our responses within.*

The manuscript presents a study on field observations of evolving snow physical properties and albedo during the early melt season on landfast ice in Canada, and uses those results for a snow albedo model analysis. The study nicely connects the changes in relevant physical properties to changes to albedo. Overall, the manuscript is well written, well organized, and with some minor revisions, it would be in good shape for publication. In particular, the discussion section was enjoyable to read. Please see comments below that I hope the authors will find useful.

*We thank the Reviewer for this positive appreciation and for constructive comments.*

**General:** It's not clear why the albedo results at the 500 nm and 1000 nm wavelengths are emphasized. Presenting the numbers in the visible (300-700 nm) and near infrared (1000+ nm) bands would make it easier to compare with previous works (e.g., Brandt et al., 2005) and be more relevant for remote sensing applications.

*This is an interesting point also made by Reviewer 1. We will not repeat our response to Reviewer 1. Briefly, our focus is on the link between snow physical properties and albedo. These are best investigated by performing calculations for specific wavelengths. We cannot extend our paper to the multiple other possible applications such as remote sensing and comparison with other works, which by the way did not measure snow SSA. In any case, our Figure 7 can be used by the interested reader to compare our data with those of Brandt et al. 2005.*

Lines 21 and 27. It would be useful to know the snow depth at which the visible band in albedo begins to decrease.

*There is no well-defined depth threshold? The snow depth at which albedo starts to decrease depends on several variables which include snow density and snow SSA. No precise value can be given. We therefore much prefer not to engage in such a discussion.*

Lines 36-38. During winter, there's little sunlight, so the albedo the surface is not important. In spring and summer, it's important.

*We agree with the Reviewer. However, this does not affect the validity of our statement, that snow in winter and spring reflects up to 90% of incoming solar radiation.*

Line 39. The melt season begins when the snow starts melting. Snow may affect the duration of sea-ice melt.

*We agree with the Reviewer. Here again, this does not affect the validity of our writing.*

Lines 40-41. This is true for thin sea ice, but not for thick sea ice. Snow has little effect on the amount of light reaching the ocean if the ice is very thick.

*Snow affects the amount of light reaching the ocean regardless office thickness. Now, we do agree with the reviewer that with specific applications in mind, ice may by itself reduce*

*incoming light sufficiently that snow effects do not matter anymore. Here, we are just mentioning a physical process.*

Line 45. The authors may be interested in reading an updated review of snow and ice optical properties by Warren: https://royalsocietypublishing.org/doi/full/10.1098/rsta.2018.0161

*Thank you for this interesting reference. It does not however affect the validity of our text.*

Line 52. It would be worthwhile to add a description in the text about the limitations of using SSA for snow crystal representation in optical modeling.

*There are sure limitations we are well aware of and we have contributed to quantify them (Picard et al., 2009). However, SSA (or equivalently optical diameter) is used in most radiative transfer models. Such a discussion would be justified in a paper focused on the importance of SSA, particle shape, etc. in optical modeling. This is however not our focus and we do not wish to lengthen our paper in what we feel are unnecessary considerations, at least for our present scope.*

Line 76. There are some cases where snow persists all summer.

*Indeed. However, this is anecdotic and does not affect our statements.*

Line 80. 'albedo drops remarkably'
It would be informative to include the albedo change from dry to melting snow. My understanding is that a change from 0.85 to 0.70 is not that remarkable relative to the change from snow (0.85) to melt ponds (0.25-0.65).

*We added "by about 0.15" line 82 to quantify this. We still think this is remarkable, as it doubles the radiative energy absorption.*

Lines 82-84. This is an overextension of results. Snow melt does not directly enhance snowfall.

*We agree with the Reviewer. Our writing very clearly indicates that that snowmelt contributes to an indirect effect that enhance snowfall. "The combined effects of surface melting and atmosphere warming enhance the air moisture content, often producing persistent overcast conditions leading to snow precipitations." This is reported in the references we cite. We never write that snow melt directly enhances snowfall.*

Lines 86-87. In some cases, the snowpack is deep enough that it never fully melts away, as observed around ridges: https://online.ucpress.edu/elementa/article/10/1/000072/169460/Spatiotemporal-evolution-of-melt-ponds-on-Arctic

*Indeed, this is the case. However, (Webster et al., 2022) themselves do write "Even so, a few snow drifts by ridges persisted throughout the melt season." Indicated that this is anecdotic with little if any impact on our arguments. We therefore feel it is not useful to lengthen our text with only marginally relevant statements.*

Lines 91-92. 'However, studies which aim to link physical and optical properties of snow still remain largely qualitative'
This isn't true. Warren, Brandt, Grenfell, Perovich, and others have made a lifetime of work in linking snow physical and optical properties, including their co-evolution. I suggest rewording this section so that it recognizes that this work is standing on the shoulder of giants and is adding to a foundation of knowledge.

*We agree that "still remain largely qualitative" is an exaggeration. What we meant is that all variables required to calculate albedo were not measured. Calculations therefore had to rely on the estimation of some physical variables, in particular snow SSA. Our study is the first to actually measure albedo on sea ice and calculate if from measured variables only, and this is the reason to focus our work on the relationship between physical and optical properties. We do recognize the extremely valuable pioneering work of Warren, Brandt, Grenfell, Perovich and others, and cite a number of their papers. However, they never measured snow SSA and we therefore add an increment to the topic. We changed "still remain largely qualitative" to "never measured all variables required to calculate albedo, and relied on the estimation of some physical variables, in particular snow SSA." Lines 93-94.*

Lines 92-93. It is true there are data limitations, but the greater limitations may be the representation of physical processes, which are difficult to appropriately incorporate as parameterisations into earth system models.

*It is not clear to us which physical processes the Reviewer is referring to.*

Lines 94-95. There are several field campaigns that have done this.

*We respectfully disagree. We are not aware of any study on sea ice which measured both optical and physical properties of snow, in particular which would have measured all snow properties required to calculate albedo, such as SSA.*

Lines 116-117. How far away was the meteorological station? It would be helpful to include that information here.

*We replaced "close to the ice camp" with "about 100 m from the ice camp". Line 119.*

Lines 135-136. What information was used to determine the auto-adjustments? Does the auto-adjustment create inconsistencies in the noise level of the measurements?

*The "auto-adjustment" consists in increasing / decreasing the integration time to maintain the maximal numerical count in the spectra (usually around 500 nm) in the range 70-100% of the maximal count (i.e. 100% is the saturation level of the sensor). This is exactly how works other field spectrometers (ASD, SVC, …). It does not create "inconsistencies" (if we understand this term correctly) because in general the integration time does not change between the incident and reflected measurements. The maximum numerical count is in fact similar for both measurements because the snow albedo is close to 1 around 500 nm. We propose to remove this sentence to avoid confusion because this is a standard operating mode, there is nothing special in this auto-adjustment.*

Line 140/Figure 2. These are useful photos. Is it possible to replace them with higher resolution versions?

*This is the TCD version, with reduced resolution. The final version will be significantly improved.*

Lines 146-147. What makes a relatively thinner snow pack less suitable? Wouldn't the combination of thin and thick be more representative?

*As stated in the following line "because thicker layers are more suitable to determine accurately irradiance profiles." Line 150. The next sentence also explains "The optical absorption by LAPs in the snowpack were determined from the exponential rate of irradiance decrease in a homogeneous layer (Belke-Brea et al., 2021; Tuzet et al., 2019) so that thicker layers yield more reliable measurements."*

Lines 177-178. The instrumental uncertainty of the probe would be helpful to include here.

*We added "(0.1°C accuracy)" line 182.*

Line 182. It would be good to expand on this a little more. What types of snow have larger uncertainties?

*We now specify "and is larger for soft snows such as depth hoar and fresh snow". Line 187*

Line 196. typo 'them'

*Thank you. Changed.*

Line 265/Figure 3. Why are there different shades for the different horizontal bars? The shades don't match the gray legend in the bottom panel.

*Throughout the Figures, light grey is for 2016 and dark grey is for 2015. This is shown in the legend box in the lower panel and in the horizontal bars separating both panels in indicating the different phases. We have slightly modified the colors and layout to insure it is obvious.*

Lines 274-275. Often, there can be melt forms near the ice-snow interface from the previous autumn. Were there no melt forms observed at the base of the snowpack?

*No melt forms at the base of the snowpack were observed. We are reporting only observations that were made, not observations that could have been made or expected and were not made, for concision. Also, please note that melt forms formed in fall can be totally transformed to depth hoar and undetectable in spring. Since we only performed spring campaigns, we feel it would be speculative and not useful to discuss all possibilities not confirmed by observations.*

Lines 279-280. It would be informative to describe how the temperature gradient was reversed. Was the temperature range the same but with the upper surface being -4.5 to -5C, or do the authors mean that the snowpack was simply warmer near the surface and cooler near the base?

*We now specify line 284 that "The subsequent increase in air temperature led to surface warming and a reversal the temperature gradient in the snowpack".*

Lines 286-287. Did snowpack temperatures increase from the top down?

*The previous paragraph about Phase I explains that at the end of Phase I the surface warmed and the temperature gradient reversed, so that the bottom of the snowpack was then colder than the top. The modification detailed above will make it clear that during phase II, temperatures decreased from the top down.*

Line 300/Figure 4. Just after the May 8 snowfall, the snow depth increases. What caused the increase if no snowfall occurred?

*Our observations started on May 13, so we unfortunately are unable to comment. The most likely response is erosion and deposition by wind, but at this point, this would be speculation so we prefer not to comment.*

Lines 315-317 and lines 320-321. I'm surprised by the higher density values for indurated depth hoar and the lower density values for wind slab in this study. Can the authors comment on this with regard to previously observed values? Is it possible that the fresh snowfall events contributed to the density measured in the uppermost portion of the snowpack, lowering the average density for the wind slab layer?

*While wind slabs are often denser that indurated depth hoar layers, it is not rare to have basal indurated depth hoar denser than upper wind slabs. We have added line 327 "Having basal indurated depth hoar denser than upper wind slabs is not rare on sea ice (Sturm et al., 2002)." Figure 3 and Table 2 of (Sturm et al., 2002) show their layer "d", indurated depth hoar, to have an average density of 344 kg m$^{-3}$, while layer "j", wind slab, has an average density of 316 kg m$^{-3}$.*

Lines 325-326. Figure 6 doesn't show the distinct vertical layers. Is there a way that this can be added to the figure?

*The Reviewer probably means horizontal layers, rather than vertical ones. Our initial draft of this Figure did feature layer boundaries and grain types. Layer boundaries have interest only if grain types are shown. However, the Figure was then so cluttered that it was essentially illegible, so we opted for the current version, which we feel is preferable for most or all readers. In many cases (not all, we admit), layer types can be inferred from SSA and density values.*

Line 330. Same comment as before that Figure 6 doesn't show the distinct vertical layers of the snowpack.

*Same response as above.*

Line 335/Figure 6. What does the white at the base of these profiles represent? Is it no data? Also, how much of the variability in the uppermost profiles before May 25 is due to spatial heterogeneity versus variable weather conditions, such as snowfall events? It may be insightful to comment on this in the text.

*White regions indeed mean no data. This has been added to the caption. We have added "Since no major modification affected the snowpack during Phase I, except near the surface, a lot of the variability observed in the stratigraphies of Figure 6 during Phase I results from spatial rather than time variability." Lines 338-339.*

Line 346. 'Some of these new layers were thick enough to be distinguishable in Figure 6.' It would be helpful to highlight these in Figure 6 somehow since they are not obvious. Was snow density of these new snowfall layers measured?

*We have specified line 358 "Some of these new layers were thick enough to be distinguishable in Figure 6 because of their high SSA." These are clearly visible on May 26, May 28, and May 30. The densities appear on the lower panel of Figure 6.*

Line 356. Typo Jun 6.

*Changed, thank you.*

Lines 357-358. This is a little confusing. How did the mass of the snow increase without notable snowfall events (Figure 4)?

*Several snowfalls did occur after June 4, as reported in Figure 4. The Reviewer may think these snowfalls are not notable because they did not result in significant increase in snow height. This however is because the general decreasing trend in snow height caused by melting masks the contribution of these snowfalls to snow height. In any case, we do not draw any strong conclusion from the impact of these snowfalls and also mention melting bottom sea ice as a cause of negative freeboard (line 365).*

Table 1 caption. Do you mean Figure 5 here? I suggest adding an additional column that describes the predominant snow layer morphology (indulated depth hoar, etc.) so that readers don't have to scroll back and forth to know which layer means what. Also, it looks like there may be a typo for Layer I.

*We do mean Figure 5, thanks for spotting the typo. We have added a column to describe the predominant snow layer morphology to Table 1. There was indeed a typo for layer I, thanks again for spotting that.*

Lines 377-378. Is it possible that the sloped surface of the dunes, and therefore the angle of reflectivity, affected the albedo measurements?

*We added lines 385-389: "The slopes of the dunes where measurements were made was barely perceptible, most likely lower than 1° in all cases. (Picard et al., 2020) have evaluated slope effects on albedo. Under diffuse light, over entirely snow-covered surfaces, a slope has no impact on albedo (their Figure 3). Considering a SZA of 45° and only direct light, the error caused by a 1° slope is slightly over 0.01. Given that the slopes involved here are <1° and that there is always significant diffuse light, we estimate that the slope-caused error is always <0.01, and in most cases much smaller, so that we neglect it."*

*In passing, the thicker snow does not necessarily mean that the surface elevation was higher. It can be that the ice is thinner and we think this is what happened in most cases.*

Line 385/Figure 7. It would be helpful to add the sample size for each panel, e.g., N = 15 to better interpret the changes between phases. It would also be informative to note what fraction of the albedo measurements were made over melt ponds.

*The sample size has been added to the caption. During the campaign pond limits were not always clear so that in most cases an ice-water mixture was measured, as now indicated in the caption.*

Line 395. Similar to the previous comment, it would be informative to note what fraction of the albedo measurements were made over melt ponds in this section.

*Same response as above.*

Line 407. 'which ranged from...'
It would be informative to add shading or thinner lines to Figure 8 to show the range or spread of the absorption spectra from the 12 samples.

*The spread is too large to allow a clear Figure. Instead we added in the caption "At 500 nm, values ranged from 0.03 to 0.42 m⁻¹" to indicate the range of values. Line 440.*

Lines 408-409. Shouldn't it be divided by the snow density to get the average absorption coefficient per volume of snow?

*Reviewer 1 made a similar comment. We again apologize for the lack of clarity. We have clarified this, lines 421-423. "It was divided by the density of water (1000 kg m⁻³) and multiplied by the average density of snow (350 kg m⁻³) in order to obtain an average absorption coefficient of the impurities in the snow."*

Lines 414-416. It's not clear how these values were determined. Was this some sort of sensitivity study with some details missing in the methods section, or manually trying different values until a decent overlap was reached with the observed average?

*To clarify this point, the new wording says "a mixture (red dashed line) of 1700 ng g⁻¹ of MD and 14.4 ng g⁻¹ of BC were found to be the best values to correctly simulates, in term of shape and amplitude, the average particulate absorption measured within the snow." Line 428.*

Line 417. These measurements were only made over snow dunes, is that correct? It would be helpful to add that note in the text here to remind readers.

*Indeed, this is written in the text, just one line down (now line 431): "we used our 2015 vertical irradiance profiles obtained in five snow dunes"*

Lines 482-483. It would be interesting to include the snow depth at which PAR becomes significant.

*We have added "below about 20 to 25 cm" line 498. This value was mentioned later on, line 571.*

Line 543. Rain events occurred? That would be informative to include in Figure 4. What impact did the rainfall have on the albedo and snow properties?

*It did rain on 22 June 2016, after all the snow was gone. The rain therefore had no impact and it is not useful to discuss it. We removed the mention to rain, line 562.*

**References**

Picard, G., Arnaud, L., Domine, F., and Fily, M.: Determining snow specific surface area from near-infrared reflectance measurements: Numerical study of the influence of grain shape, Cold Regions Sci. Tech., 56, 10-17, 10.1016/j.coldregions.2008.10.001, 2009.

Sturm, M., Holmgren, J., and Perovich, D. K.: Winter snow cover on the sea ice of the Arctic Ocean at the Surface Heat Budget of the Arctic Ocean (SHEBA): Temporal evolution and spatial variability, J. Geophys. Res., 107, 8047, 10.1029/2000jc000400, 2002.

Webster, M. A., Holland, M., Wright, N. C., Hendricks, S., Hutter, N., Itkin, P., Light, B., Linhardt, F., Perovich, D. K., Raphael, I. A., Smith, M. M., von Albedyll, L., and Zhang, J.: Spatiotemporal evolution of melt ponds on Arctic sea ice: MOSAiC observations and model results, Elementa: Science of the Anthropocene, 10, 10.1525/elementa.2021.000072, 2022.